# LEARNING TO LOCALIZE LEAKAGE OF CRYPTO-GRAPHIC KEYS THROUGH POWER CONSUMPTION

## ABSTRACT

While cryptographic algorithms such as the ubiquitous Advanced Encryption Standard (AES) are secure, *physical implementations* of these algorithms in hardware inevitably 'leak' sensitive information such as cryptographic keys. A particularly insidious form of leakage arises from the fact that hardware's power consumption over time is statistically associated with the data it processes and the instructions it executes. Supervised deep learning has emerged as a state-of-the-art tool for carrying out *power side-channel attacks*, which exploit this leakage to break cryptographic implementations by learning to map power consumption measurements recorded during encryption to the secret key used for that encryption. In this work, we seek instead to develop a principled deep learning framework for *defense* against such attacks by understanding the relative leakage due to power measurements recorded at different points in time. This information is invaluable to cryptographic hardware designers for understanding *why* their hardware leaks and how they can mitigate the leakage (e.g. by indicating that a particular section of code or electronic component is responsible for leakage and should be revised). Towards this end, we propose a novel deep learning algorithm by formulating an adversarial game played between a classifier trained to estimate the conditional distribution of a key given power measurements, and an 'obfuscator' which probabilistically erases individual power measurements and is trained to minimize the classifier-estimated log-likelihood of the correct key, subject to a penalty on erasure probability. We theoretically characterize the ideal output of our algorithm in terms of conditional mutual information quantities involving the key and individual power measurements. We then empirically demonstrate the efficacy of our algorithm on real and synthetic datasets of power measurements from implementations of the AES cryptographic standard. Our code can be found in the supplementary materials.

## 1 INTRODUCTION

The Advanced Encryption Standard (AES) (Daemem & Rijmen, 1999; Daemen & Rijmen, 2013) is widely used and trusted for protecting sensitive data. For example, it is approved by the United States National Security Agency for protecting top secret information (Committee on National Security Systems, 2003), it is a major component of the Transport Layer Security (TLS) protocol (Rescorla, 2000) which underlies the security of HTTPS (Rescorla, 2000), and is used in payment card readers to secure card information before transmission to financial institutions (Bluefin Payment Systems, 2023).

AES aims to keep data secret when it is transmitted over insecure channels that are accessible to unknown and untrusted parties (e.g. via wireless transmissions which may be intercepted, or storage on hard drives which may be accessed by untrusted individuals). Prior to transmission, the data is first encoded and partitioned into a sequence of fixed-length bitstrings called *plaintexts*. Each plaintext is then *encrypted* into a *ciphertext* by applying an invertible function from a family of functions indexed by an integer called a *cryptographic key*. This family of functions is designed so that if the key is sampled uniformly at random, then the plaintext and ciphertext are marginally independent. The key is known to the sender and intended recipients of the transmission,[1] and is kept secret from potential

---

[1]For example, the key may be exchanged and periodically updated using an asymmetric-key cryptographic algorithm such as RSA or Diffie-Hellman (Paar & Pelzl, 2010, ch. 6) Such algorithms do not require the sender

eavesdroppers. Thus, the intended recipients can use the key to *decrypt* the ciphertext back into the original plaintext, while eavesdroppers who possess the ciphertext but not the key learn nothing about the plaintext.

Clearly, such an algorithm is effective only if the cryptographic key remains outside of the hands of eavesdroppers. AES is believed to be 'algorithmically secure' in the sense that given an AES implementation with a fixed key, it is not feasible to determine the key by encrypting a chosen sequence of plaintexts and observing the resulting ciphertexts (Mouha, 2021). For reference, to our knowledge, the best known attack on the 128-bit version of AES under realistic conditions would require about $2^{125}$ such encryptions on average to successfully determine the key (Tao & Wu, 2015), compared to $2^{127}$ encryptions for a naive brute-force attack which randomly guesses and checks keys until success.

Despite the 'algorithmic' security of AES and other cryptographic algorithms, *physical implementations* of these algorithms in hardware inevitably 'leak' information about their cryptographic keys. This phenomenon, called *side-channel leakage*, occurs because hardware produces measurable physical signals that are statistically associated with the data it processes and the instructions it executes. In this work, we consider *power side-channel leakage*, i.e. statistical association between a device's cryptographic key and its power consumption over time while encrypting data, which is a major security vulnerability for AES implementations (Kocher et al., 1999; Bronchain & Standaert, 2020). Note, however, that hardware emits many diverse physical signals which cause side-channel leakage, such as electromagnetic radiation (Quisquater & Samyde, 2001; Genkin et al., 2016), program/operation execution time (Kocher, 1996; Lipp et al., 2018; Kocher et al., 2019), and sound due to vibration of electronic components (Genkin et al., 2014). Refer to appendix A for some simple intuition-building examples of side-channel leakage.

Cryptographic implementations can be circumvented by *side-channel attacks*, which exploit side-channel leakage to learn the cryptographic key of some target device. In this work, we consider *profiling* power side-channel attacks, in which the attacker is assumed to possess a clone of the target device and can repeatedly measure its power consumption over time while encrypting arbitrary plaintexts using arbitrary keys. Measured power consumption during encryption is encoded as a real vector called a *power trace*, where each element encodes the power measurement at a fixed point in time relative to the start of encryption. Attackers can use the clone device to model the conditional distribution of the cryptographic key given the power trace, and can then collect power traces from the target device and identify the key which maximizes the likelihood of the key and power traces according to their model.

Supervised deep learning has emerged as a state-of-the-art technique for carrying out profiled power side-channel attacks, achieving comparable or superior performance to prior approaches with far less data preprocessing and feature selection (Maghrebi et al., 2016; Benadjila et al., 2020; Zaid et al., 2020; Wouters et al., 2020; Bursztein et al., 2023). The limitations of classical (non-deep learning) attacks include assuming specific forms for the conditional distribution of keys given traces (Chari et al., 2003; Schindler et al., 2005; Hospodar et al., 2011), requiring feature selection or principal component analysis to significantly reduce the dimensionality of power traces (Chari et al., 2003; Archambeau et al., 2006), and limited ability to exploit $n$-th order leakage where $n > 1$, i.e. where the key is dependent on a set of $n$ measurements but independent of all of its subsets with cardinality less than $n$ (Messerges, 2000; Agrawal et al., 2005). In contrast, neural nets are asymptotically universal function approximators, and hence can in principle represent nearly arbitrary conditional distributions (Hornik et al., 1989). CNNs and transformers have proven capable of operating on raw power traces without feature selection or dimensionality reduction (Lu et al., 2021; Bursztein et al., 2023), and neural nets are effective against 'masking' countermeasures which exploit the difficulty of exploiting 2nd-order leakage with classical attacks (Benadjila et al., 2020; Zaid et al., 2020; Wouters et al., 2020). Thus, deep learning is a major threat to a wide assortment of security measures and evaluations that were designed with the limitations of classical attacks in mind.

In this work, we seek to instead leverage deep learning to *defend* against side-channel attacks by identifying specific points in time at which power consumption is 'useful' for predicting the key. Our intent is to enable the designers of implementations to understand why their implementations leak (e.g. which machine instructions are responsible, whether their countermeasures are effective), as

---

and recipient to know the same secret key, but are slower than AES, so it is common to use them only for key exchange and to use AES for transmission of large quantities of data.

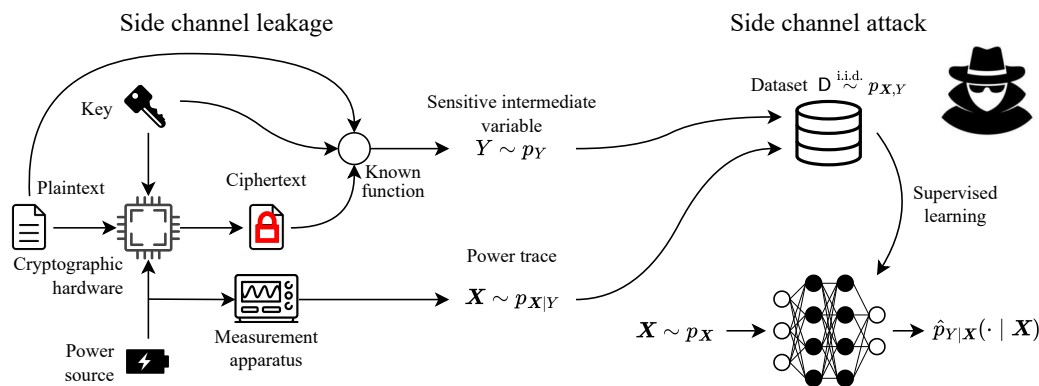

Figure 1: Diagram illustrating our probabilistic framing of power side-channel leakage. AES hardware encrypts a plaintext given a key, resulting in a ciphertext. The power consumption over time of the hardware is measured during encryption and encoded as a vector called a power trace. Consider a 'sensitive' intermediate variable in the cryptographic algorithm which is a known function of the key, plaintext, and ciphertext, and which gives information about the key given the plaintext and ciphertext. We view the power trace and sensitive variable as realizations of jointly-distributed random variables $\boldsymbol{X}, Y \sim p_{\boldsymbol{X},Y}$ respectively, and side channel attacks can be carried out by using supervised learning to estimate $p_{Y|\boldsymbol{X}}$.

opposed to a mere indication of how vulnerable the implementation is to attacks. Towards this end, we propose a novel algorithm that localizes leakage using an adversarial game played between a deep neural network classifier trained to map power traces to the associated AES keys (via a 'sensitive' intermediate variable), and a data-independent 'obfuscator' which randomly erases individual power measurements in these traces and is trained to maximize the loss of the classifier, subject to a penalty on the erasure probabilities. The optimal erasure probabilities constitute a trade-off between being large and depriving the classifier of the information provided by the associated power measurement, and being small to reduce the penalty. Thus, after training, measurements that have 'high leakage' in the sense of being more useful to the classifier will have high erasure probabilities, and 'low leakage' measurements will have low erasure probabilities, so the probabilities can be observed to identify the relative leakage of different measurements. Our key contributions are as follows:

- We propose the aforementioned novel deep learning-based power side-channel leakage localization algorithm.

- We theoretically characterize the ideal solution of our algorithm's optimization problem in terms of conditional mutual information quantities involving the cryptographic key and individual power measurements.

- We experimentally demonstrate the efficacy of our algorithm using simulated and real power side-channel attack datasets. To facilitate the latter, we further propose a novel performance metric for evaluating the correctness of an attempt to localize leakage when we lack ground truth knowledge about which power measurements are leaking.

## 2 BACKGROUND AND SETTING

Here we provide a probabilistic framing of power side-channel leakage. Refer to appendix A for a more detailed background which includes simple intuition-building examples of side-channel leakage.

### 2.1 PROBABILISTIC FRAMING OF POWER SIDE-CHANNEL LEAKAGE

See figure 1 for a diagram illustrating our setting. We assume to have a symmetric-key cryptographic device that encrypts data in a manner dependent on some sensitive intermediate variable $y \in \mathsf{Y}$, where $\mathsf{Y}$ is a finite set (e.g. consisting of bytestrings encoding all possible values of the variable). We assume to have some measurement apparatus that allows us to measure power traces during encryption, encoded as $\boldsymbol{x} \in \mathbb{R}^T$ where $T \in \mathbb{Z}_{++}$ denotes the number of measurements recorded per

encryption. We view the power traces and sensitive variables as realizations of jointly-distributed random variables $\boldsymbol{X}, Y \sim p_{\boldsymbol{X},Y}$ respectively, where $p_Y$ is a simple known distribution (e.g. uniform) and $p_{\boldsymbol{X}|Y}$ is *a priori* unknown and dictated by factors such as the hardware, environment, and measurement setup. In this work we assume that the conditional density functions $p_{\boldsymbol{X}|Y}(\cdot \mid y)$ exist and have support equal to $\mathbb{R}^T$, which is reasonable because power consumption usually has a 'random' component which is well-described by additive Gaussian noise (Mangard et al., 2007). This setting is amenable to supervised learning, and most profiled power side channel attacks (e.g. based on deep learning) are based on collecting a dataset $\mathsf{D} \overset{\text{i.i.d.}}{\sim} p_{\boldsymbol{X},Y}$ and using it to model the conditional distribution $p_{Y|\boldsymbol{X}}$.

## 2.2 Quantifying side-channel leakage using conditional mutual information

Given that devices are vulnerable to side-channel attacks because of the statistical association between $\boldsymbol{X}$ and $Y$, it is reasonable to quantify the amount of leakage at a timestep $t \in [1 .. T]$ using Shannon (conditional) mutual information (Shannon, 1948) between $X_t$ and $Y$:

$$\mathbb{I}[Y; X_t \mid \mathsf{S}] := \mathbb{E}\left[\log p_{Y|X_t,\mathsf{S}}(Y \mid X_t, \mathsf{S}) - \log p_{Y|\mathsf{S}}(Y \mid \mathsf{S})\right] \tag{1}$$

where $\mathsf{S} \subset \{X_1, \ldots, X_T\} \setminus \{X_t\}$. Intuitively, this quantity tells us the extent to which our uncertainty about $Y$ is reduced upon observing $X_t$, provided we have already observed the elements of $\mathsf{S}$.

Clearly, we should consider timestep $t$ to be leaking if $X_t$ is directly associated with $Y$. For example, consider the device characterized by Mangard et al. (2007) which consumes power in proportion to the Hamming weight (number of nonzero bits) of the chunk of data it is presently operating on, and suppose our chosen sensitive variable is the output of the first SubBytes operation, as is common when attacking AES implementations (see appendix A for elaboration). In this case, if the device carries out the first SubBytes operation at time $t$, we expect to have $\mathbb{I}[Y; X_t] > 0$.

Intuitively, it is very reasonable to consider timestep $t$ to be leaking if $\mathbb{I}[Y; X_t] > 0$. More subtle is the fact that $\mathbb{I}[Y; X_t] = 0$ *does not imply* that $X_t$ is 'innocuous'. It may be the case that $X_t$ tells us nothing about $Y$ by itself, but it tells us something useful *in combination with* some $X_\tau$ for $\tau \neq t$. For example, the power consumption of electronic devices has 'inertia' due to fundamental physical laws and intentional design decisions. Suppose our aforementioned device carries out the first SubBytes operation at time $t + 1$. Due to this 'inertia', $X_{t+1}$ will depend not only on $Y$, but also on $X_t$. Thus, even if $X_t$ is independent of $Y$, $X_t$ is *dependent* on $Y$ given $X_{t+1}$. This is because by learning $X_t$, we can 'subtract' its influence from $X_{t+1}$, thereby isolating the component of $X_{t+1}$ which depends on $Y$. In this case, we expect that $\mathbb{I}[Y; X_t] = 0$ but $\mathbb{I}[Y; X_t \mid X_{t+1}] > 0$. In general, to have a reasonable notion of leakage at time $t$, we must consider not only $\mathbb{I}[Y; X_t]$, but $\mathbb{I}[Y; X_t \mid \mathsf{S}]$ for every $\mathsf{S} \subset \{X_1, \ldots, X_T\} \setminus \{X_t\}$.

## 3 Method: Adversarial Leakage Localization (ALL)

Given $\boldsymbol{X}, Y \sim p_{\boldsymbol{X},Y}$ as defined above with $\boldsymbol{X} = (X_t : t = 1, \ldots, T)$, we seek to assign to each $t \in [1 .. T]$ a scalar indicating the 'amount of leakage' of information about $Y$ due to power measurement $X_t$. Clearly the quantities $\{\mathbb{I}[Y; X_t \mid \mathsf{S}] : \mathsf{S} \subset \{X_1, \ldots, X_T\} \setminus \{X_t\}\}$ give us insight into leakage, but it is not obvious how they should be weighted in a single scalar leakage measurement. In this section, we will propose an optimization problem with a solution that can be interpreted as a leakage measurement assignment to each timestep according to an intuitively reasonable weighting scheme. We will then prove that given ideal assumptions, its solution is equivalent to that of an adversarial game played between a neural network classifier trained to map realizations of $\boldsymbol{X}$ to $Y$ where some of the elements of $\boldsymbol{X}$ have been 'erased', and an 'obfuscator' which probabilistically erases elements of these realizations of $\boldsymbol{X}$ in order to maximize the loss of the classifier, subject to a penalty on erasure probability. Refer to appendix B for an extended version of this section with proofs and additional results.

## 3.1 Optimization problem

We define a vector $\boldsymbol{\gamma} \in [0,1]^T$ which we name the *erasure probabilities*. This vector will parameterize a distribution over binary vectors in $\{0,1\}^T$ as follows:

$$\boldsymbol{\mathcal{A}}_{\boldsymbol{\gamma}} \sim p_{\boldsymbol{\mathcal{A}}_{\boldsymbol{\gamma}}} \quad \text{where} \quad \mathcal{A}_{\boldsymbol{\gamma},t} = \begin{cases} 1 & \text{with probability} \quad 1 - \gamma_t \\ 0 & \text{with probability} \quad \gamma_t, \end{cases} \tag{2}$$

i.e. its elements are independent Bernoulli random variables where the $t$-th element has parameter $p = 1 - \gamma_t$. For arbitrary vectors $\boldsymbol{x} \in \mathbb{R}^T$, $\boldsymbol{\alpha} \in \{0,1\}^T$, let us denote $\boldsymbol{x}_{\boldsymbol{\alpha}} := (x_t : t = 1, \ldots, T : \alpha_t = 1)$, i.e. the sub-vector of $\boldsymbol{x}$ containing its elements for which the corresponding element of $\boldsymbol{\alpha}$ is 1. We can accordingly use $\boldsymbol{\mathcal{A}}_{\boldsymbol{\gamma}}$ to obtain random sub-vectors $\boldsymbol{X}_{\boldsymbol{\mathcal{A}}_{\boldsymbol{\gamma}}}$ of $\boldsymbol{X}$. Note that $\gamma_t$ represents the probability that $X_t$ will *not* be an element of $\boldsymbol{X}_{\boldsymbol{\mathcal{A}}_{\boldsymbol{\gamma}}}$ (thus, 'erasure probability').

Consider the optimization problem

$$\min_{\boldsymbol{\gamma} \in [0,1]^T} \quad \mathcal{L}_{\text{ideal}}(\boldsymbol{\gamma}) := \frac{1}{2}\lambda \|\boldsymbol{\gamma}\|_2^2 + \mathbb{I}[Y; \boldsymbol{X}_{\boldsymbol{\mathcal{A}}_{\boldsymbol{\gamma}}} \mid \boldsymbol{\mathcal{A}}_{\boldsymbol{\gamma}}]. \tag{3}$$

where $\lambda > 0$ is a hyperparameter. Intuitively, this is a trade-off between having high erasure probabilities and thereby reducing the mutual information quantity, and having low erasure probabilities to reduce the norm penalty. For each $t$, we can write

$$\mathcal{L}_{\text{ideal}}(\boldsymbol{\gamma}) = \frac{1}{2}\lambda\gamma_t^2 + \sum_{\boldsymbol{\alpha} \in \{0,1\}^T : \, \alpha_t = 0} [\mathbb{I}[Y; X_t, \boldsymbol{X}_{\boldsymbol{\alpha}}] - \gamma_t\, \mathbb{I}[Y; X_t \mid \boldsymbol{X}_{\boldsymbol{\alpha}}]]\, p_{\boldsymbol{\mathcal{A}}_{\boldsymbol{\gamma},-t}}(\boldsymbol{\alpha}_{-t}) + \frac{1}{2}\lambda \|\boldsymbol{\gamma}_{-t}\|_2^2. \tag{4}$$

As $\gamma_t$ decreases, so does the penalty $\frac{1}{2}\lambda\gamma_t^2$. As $\gamma_t$ increases, the quantities $-\gamma_t\, \mathbb{I}[Y; X_t \mid \boldsymbol{X}_{\boldsymbol{\alpha}}]$ decrease in proportion to $\mathbb{I}[Y; X_t \mid \boldsymbol{X}_{\boldsymbol{\alpha}}]$. For $\boldsymbol{\gamma}^* \in \arg\min_{\boldsymbol{\gamma} \in [0,1]^T} \mathcal{L}_{\text{ideal}}(\boldsymbol{\gamma})$, it follows that $\gamma_t^*$ positively correlates with some notion of a 'typical' value of $\mathbb{I}[Y; X_t \mid \boldsymbol{X}_{\boldsymbol{\alpha}}]$, and it appears reasonable to view it as a notion of the 'amount of leakage' at time $t$.

We can verify that a solution to equation 3, and derive an implicit expression for it:

**Proposition 1.** *For $\mathcal{L}_{ideal}$ as defined in equation 3, $\arg\min_{\boldsymbol{\gamma} \in [0,1]^T} \mathcal{L}_{ideal}(\boldsymbol{\gamma}) \neq \emptyset$. Furthermore, every $\boldsymbol{\gamma}^* \in \arg\min_{\boldsymbol{\gamma} \in [0,1]^T} \mathcal{L}_{ideal}(\boldsymbol{\gamma})$ must satisfy*

$$\gamma_t^* = \min\left\{ \frac{1}{\lambda} \sum_{\boldsymbol{\alpha} \in \{0,1\}^T : \, \alpha_t = 0} \mathbb{I}[Y; X_t \mid \boldsymbol{X}_{\boldsymbol{\alpha}}] \prod_{\tau \in [1 \, .. \, T] \setminus \{t\}} (\gamma_\tau^*)^{1-\alpha_\tau}(1-\gamma_\tau^*)^{\alpha_\tau}, \quad 1 \right\} \quad \forall t \in [1 \, .. \, T]. \tag{5}$$

*Sketch of proof (full proof).* The existence of a solution follows from the extreme value theorem because $[0,1]^T$ is compact and $\mathcal{L}_{\text{ideal}}(\boldsymbol{\gamma})$ is continuous in $\boldsymbol{\gamma}$. We derive the expression for $\gamma_t^*$ by expressing our objective function as $f_1(\boldsymbol{\gamma}_{-t}) + \gamma_t f_2(\boldsymbol{\gamma}_{-t}) + \frac{1}{2}\lambda\gamma_t^2$ and computing the first and second partial derivatives with respect to $\gamma_t$. The first partial derivative always has a zero, which may or may not be feasible. The second partial derivative is equal to $\lambda > 0$. Thus, if the zero is feasible, it is the solution. If it is not feasible, we show that the first partial derivative is negative for all $\gamma_t \in [0,1]$, implying that the objective is minimized for $\gamma_t = 1$. $\qquad\square$

**Corrolary 1.1.** *Under the conditions of Proposition 1, if $\lambda > \log|\mathsf{Y}|$, then $\boldsymbol{\gamma}^* \in [0,1)^T$.*

*Sketch of proof (full proof).* This follows from noting that each conditional mutual information term is upper-bounded by the Shannon entropy of $Y$, which in turn is upper-bounded by the Shannon entropy of a uniform distribution over $\mathsf{Y}$. $\qquad\square$

This suggests that we should choose $\lambda$ to be large enough that no $\gamma_t^*$ saturates at 1, and that for consistency we should quantify leakage using $\lambda\boldsymbol{\gamma}^*$ rather than $\boldsymbol{\gamma}^*$.

## 3.2 EQUIVALENT ADVERSARIAL GAME

In practice we cannot directly solve equation 3 because we lack an expression for $p_{\boldsymbol{X},Y}$. Here we will propose a different optimization problem which is equivalent to equation 3 given ideal assumptions, and allows us to use deep learning to characterize $p_{\boldsymbol{X},Y}$ using data.

Consider the family $\Phi := \{\Phi_{\boldsymbol{\alpha}}\}_{\boldsymbol{\alpha} \in \{0,1\}^T}$ with each element a deep neural network

$$\Phi_{\boldsymbol{\alpha}} : \mathsf{Y} \times \mathbb{R}^{\sum_{t=1}^T \alpha_t} \times \mathbb{R}^P \to \mathbb{R}_+ : (y, \boldsymbol{x}, \boldsymbol{\theta}) \mapsto \Phi_{\boldsymbol{\alpha}}(y \mid \boldsymbol{x}; \boldsymbol{\theta}). \tag{6}$$

We denote by $\Phi_{\boldsymbol{\alpha}}(y \mid \boldsymbol{x}; \boldsymbol{\theta})$ the mass assigned to $y$ by the network $\Phi_{\boldsymbol{\alpha}}$ with weights $\boldsymbol{\theta}$ and input $\boldsymbol{x}$. We assume that each $\Phi_{\boldsymbol{\alpha}}(\cdot \mid \boldsymbol{x}; \boldsymbol{\theta})$ is a probability mass function over $\mathsf{Y}$ (e.g. the neural net has a softmax output activation). Consider the optimization problem

$$\min_{\boldsymbol{\gamma} \in [0,1]^T} \max_{\boldsymbol{\theta} \in \mathbb{R}^P} \quad \mathcal{L}_{\text{adv}}(\boldsymbol{\gamma}, \boldsymbol{\theta}) := \frac{1}{2}\lambda \|\boldsymbol{\gamma}\|_2^2 + \mathbb{E} \log \Phi_{\boldsymbol{\mathcal{A}}_{\boldsymbol{\gamma}}}(Y \mid \boldsymbol{X}_{\boldsymbol{\mathcal{A}}_{\boldsymbol{\gamma}}}; \boldsymbol{\theta}). \tag{7}$$

**Proposition 2.** *Consider the objective function $\mathcal{L}_{adv}$ of equation 7. Suppose there exists some $\boldsymbol{\theta}^* \in \mathbb{R}^P$ such that $\Phi_{\boldsymbol{\alpha}}(y \mid \boldsymbol{x}_{\boldsymbol{\alpha}}; \boldsymbol{\theta}^*) = p_{Y \mid \boldsymbol{X}_{\boldsymbol{\alpha}}}(y \mid \boldsymbol{x}_{\boldsymbol{\alpha}})$ for all $\boldsymbol{\alpha} \in \{0,1\}^T$, $\boldsymbol{x} \in \mathbb{R}^T$, $y \in \mathsf{Y}$. Then*

$$\boldsymbol{\theta}^* \in \arg\max_{\boldsymbol{\theta} \in \mathbb{R}^P} \mathcal{L}_{adv}(\boldsymbol{\gamma}, \boldsymbol{\theta}) \quad \forall \boldsymbol{\gamma} \in [0,1]^T. \tag{8}$$

*Furthermore, for all $y \in \mathsf{Y}$ and for all $\boldsymbol{\gamma} \in [0,1]^T$, $\boldsymbol{\alpha} \in \{0,1\}^T$ such that $p_{\boldsymbol{\mathcal{A}}_{\boldsymbol{\gamma}}}(\boldsymbol{\alpha}) > 0$,*

$$\Phi_{\boldsymbol{\alpha}}(y \mid \boldsymbol{X}_{\boldsymbol{\alpha}}; \hat{\boldsymbol{\theta}}) = p_{Y \mid \boldsymbol{X}_{\boldsymbol{\alpha}}}(y \mid \boldsymbol{X}_{\boldsymbol{\alpha}}) \quad p_{\boldsymbol{X}}\text{-almost surely} \quad \forall \hat{\boldsymbol{\theta}} \in \arg\min_{\boldsymbol{\theta} \in \mathbb{R}^P} \mathcal{L}_{adv}(\boldsymbol{\gamma}, \boldsymbol{\theta}). \tag{9}$$

*Sketch of proof (full proof).* The first claim follows straightforwardly from Gibbs' inequality. The second claim follows from re-writing the difference $\mathcal{L}_{\text{adv}}(\boldsymbol{\gamma}, \boldsymbol{\theta}^*) - \mathcal{L}_{\text{adv}}(\boldsymbol{\gamma}, \hat{\boldsymbol{\theta}})$ as a function of KL divergences between distributions $p_{Y \mid \boldsymbol{X}_{\boldsymbol{\alpha}}}(\cdot \mid \boldsymbol{x}_{\boldsymbol{\alpha}})$ and $\Phi_{\boldsymbol{\alpha}}(\cdot \mid \boldsymbol{x}_{\boldsymbol{\alpha}}; \hat{\boldsymbol{\theta}})$, which makes it clear that the difference is nonnegative and equal to zero if and only if the claim is satisfied. $\qquad\square$

**Corrolary 2.1.** *Under the assumptions of Proposition 2, equations 7 and 3 are equivalent.*

*Sketch of proof (full proof).* We first note that for any $\hat{\boldsymbol{\theta}} \in \arg\max_{\boldsymbol{\theta} \in \mathbb{R}^P} \mathcal{L}_{\text{adv}}(\boldsymbol{\gamma}, \boldsymbol{\theta})$, we can replace each $\Phi_{\boldsymbol{\alpha}}(y \mid \boldsymbol{x}_{\boldsymbol{\alpha}}; \hat{\boldsymbol{\theta}})$ by $p_{Y \mid \boldsymbol{X}_{\boldsymbol{\alpha}}}(y \mid \boldsymbol{x}_{\boldsymbol{\alpha}})$ in $\min_{\boldsymbol{\gamma} \in [0,1]^T} \mathcal{L}_{\text{adv}}(\boldsymbol{\gamma}, \hat{\boldsymbol{\theta}})$ without changing its solution. We then algebraically manipulate $\min_{\boldsymbol{\gamma} \in [0,1]^T} \mathcal{L}_{\text{adv}}(\boldsymbol{\gamma}, \hat{\boldsymbol{\theta}})$ using information theoretic identities and dropping additive constants which do not depend on $\boldsymbol{\gamma}$ until we arrive at equation 3. $\qquad\square$

## 3.3 IMPLEMENTATION DETAILS

It would be impractical to train $2^T$ neural networks independently, so we amortize the cost by instead training a single network with $\boldsymbol{\alpha}$ as an auxiliary input:

$$\Phi : \mathsf{Y} \times \mathbb{R}^T \times \{0,1\}^T \times \mathbb{R}^P : (y, \tilde{\boldsymbol{x}}, \boldsymbol{\alpha}, \boldsymbol{\theta}) \mapsto \Phi(y \mid \tilde{\boldsymbol{x}}, \boldsymbol{\alpha}, \boldsymbol{\theta}) \tag{10}$$

where each $\Phi_{\boldsymbol{\alpha}}(y \mid \boldsymbol{x}_{\boldsymbol{\alpha}}; \boldsymbol{\theta}) := \Phi(y \mid \boldsymbol{\alpha} \odot \boldsymbol{x}, \boldsymbol{\alpha}; \boldsymbol{\theta})$. We can then re-write equation 7 as

$$\min_{\boldsymbol{\gamma} \in [0,1]^T} \max_{\boldsymbol{\theta} \in \mathbb{R}^P} \quad \mathcal{L}(\boldsymbol{\gamma}, \boldsymbol{\theta}) := \frac{1}{2}\lambda \|\boldsymbol{\gamma}\|_2^2 + \mathbb{E} \log \Phi(Y \mid \boldsymbol{\mathcal{A}}_{\boldsymbol{\gamma}} \odot \boldsymbol{X}, \boldsymbol{\mathcal{A}}_{\boldsymbol{\gamma}}; \boldsymbol{\theta}). \tag{11}$$

See figure 2 for an illustration of this implementation. We intent to approximately solve equation 11 using an alternating minibatch stochastic gradient descent-style technique, similarly to GANs (Goodfellow et al., 2014). To do so, we must first convert it into an equivalent unconstrained optimization problem so that it is amenable to gradient descent. We must then derive expressions for the gradients of our objective function which can be estimated using automatic differentiation (Paszke et al., 2019) and Monte Carlo integration.

Note that the inner optimization problem is already unconstrained, and it is immediate that

$$\nabla_{\boldsymbol{\theta}} \mathcal{L}(\boldsymbol{\gamma}, \boldsymbol{\theta}) = \mathbb{E} \nabla_{\boldsymbol{\theta}} \log \Phi(Y \mid \boldsymbol{\mathcal{A}}_{\boldsymbol{\gamma}} \odot \boldsymbol{X}, \boldsymbol{\mathcal{A}}_{\boldsymbol{\gamma}}; \boldsymbol{\theta}). \tag{12}$$

We can re-parameterize the outer problem and derive an appropriate gradient expression as follows:

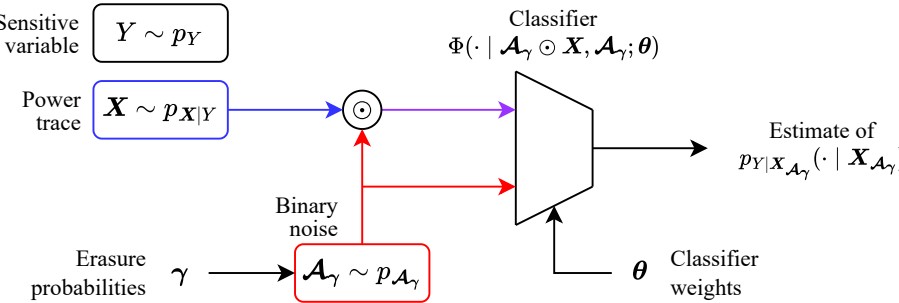

Figure 2: A diagram illustrating our adversarial leakage localization technique. We sample a sensitive variable $Y \sim p_Y$ and a power trace $X \sim p_{X|Y}$ from our training dataset, and a binary noise vector $\mathcal{A}_\gamma \sim p_{\mathcal{A}_\gamma}$ such that each element at position $t$ is equal to 0 with probability $\gamma_t$ and 1 with probability $1 - \gamma_t$, where $\gamma \in [0, 1]^T$ denotes our obfuscation weights. We multiply the power trace elementwise by the binary noise vector, and feed both the noisy power trace $\mathcal{A}_\gamma \odot X$ and the noise vector $\mathcal{A}_\gamma$ as inputs to our classifier $\Phi$. The classifier is trained to maximize the log-likelihood of $Y$ given $X_{\mathcal{A}_\gamma}$, while simultaneously the erasure probabilities are trained to minimize this log-likelihood, subject to a norm penalty which pushes them towards $\mathbf{0}$.

**Proposition 3.** *The optimization problem of equation 11 is equivalent to*

$$\min_{\gamma' \in (\mathbb{R} \cup \{\pm\infty\})^T} \max_{\theta \in \mathbb{R}^P} \quad \mathcal{L}(\mathrm{Sigmoid}(\gamma'), \theta) \tag{13}$$

*where $\gamma = \mathrm{Sigmoid}(\gamma')$. Furthermore, we can express*

$$\nabla_{\gamma'} \mathcal{L}(\gamma, \theta) = \lambda \gamma \odot \gamma \odot \overline{\gamma} + \mathbb{E}\left[\log \Phi(Y \mid \mathcal{A}_\gamma \odot X, \mathcal{A}_\gamma; \theta) \cdot \left(\overline{\mathcal{A}}_\gamma \odot \overline{\gamma} - \mathcal{A}_\gamma \odot \gamma\right)\right], \tag{14}$$

*where for compactness we have left implicit that $\gamma$ is a function of $\gamma'$, and have denoted $\overline{\mathcal{A}}_\gamma := \mathbf{1} - \mathcal{A}_\gamma$ and $\overline{\gamma} := \mathbf{1} - \gamma$.*

*Sketch of proof (full proof).* The re-parameterization is valid because $\mathrm{Sigmoid}$ is bijective and $\mathrm{Sigmoid}\left((\mathbb{R} \cup \{\pm\infty\})^T\right) = [0, 1]^T$. When deriving the expression for the gradient, we cannot immediately exchange the order of expectation and differentiation because the expectation is taken over $\mathcal{A}_\gamma$ where $p_{\mathcal{A}_\gamma}$ depends on $\gamma'$. Instead, we use the REINFORCE estimator (Williams, 1992) and simplify the expression using the definition of $p_{\mathcal{A}_\gamma}$. $\square$

See algorithm 1 for pseudo-code describing a simplified algorithm to approximately solve equation 11. Additional engineering details and performance-enhancing tweaks can be found in algorithm 3 of appendix B. Note that a straightforward implementation of our algorithm would update $\theta$ to minimize $\mathcal{L}(\mathrm{Sigmoid}(\gamma'), \theta)$. However, in practice we find that this version of the algorithm is highly sensitive to $\lambda$. Choosing $\lambda$ too small will lead to elements of $\gamma$ saturating at 1, and choosing it too large will push all elements of $\gamma$ close to zero, resulting in the classifier training almost exclusively on the inputs $(\mathbf{1} \odot x, \mathbf{1})$. However, Proposition 2 implies that in the ideal case, our algorithm is equivalent to equation 3 when the classifier is trained with noise sampled from *any* full-support distribution over $\{0, 1\}^T$, not just $p_{\mathcal{A}_\gamma}$. We thus train our classifier with noise sampled from $\mathcal{U}(\{0, 1\}^T)$, and find that this version of the algorithm is significantly easier to tune.

## 4 RELATED WORK

The adversarial nature of our algorithm was inspired by GANs (Goodfellow et al., 2014). The use of classifiers with probabilistic input ablation for mutual information estimation was inspired by the causal graph edge detection technique of the ENCO algorithm (Lippe et al., 2022). However, due to the distinct nature of our problem, our algorithm departs significantly from both of these.

A great deal of prior work has applied neural net interpretability techniques for tasks similar to power side channel leakage localization (Masure et al., 2019; Hettwer et al., 2020; Jin et al., 2020; Wouters et al., 2020; Zaid et al., 2020; van der Valk et al., 2021; Wu et al., 2021; Golder et al., 2022; Li et al.,

---

**Algorithm 1:** A practical implementation of our algorithm, simplified for clarity.

**Input:** Dataset $D := \{(\boldsymbol{x}^{(n)}, y^{(n)}) : n \in [1 .. N]\} \subset \mathbb{R}^T \times \mathsf{Y}$, initial classifier weights $\boldsymbol{\theta}^{(0)} \in \mathbb{R}^P$, initial unsquashed erasure probabilities $\boldsymbol{\gamma}'^{(0)} \in \mathbb{R}^T$, norm penalty coefficient $\lambda \in \mathbb{R}_+$

**Output:** Trained parameters $\hat{\boldsymbol{\theta}} \in \mathbb{R}^P, \hat{\boldsymbol{\gamma}}' \in \mathbb{R}^T$

1   $t \leftarrow 0$          `// training step counter`
2   **while** *not converged* **do**
3      Choose $n \in [1 .. N]$        `// datapoint index`
4      $\boldsymbol{\alpha}^{(t)} \sim \mathcal{U}(\{0,1\}^T)$        `// binary noise`
5      $l^{(t)} \leftarrow \log \Phi(y^{(n)} \mid \boldsymbol{\alpha}^{(t)} \odot \boldsymbol{x}^{(n)}, \boldsymbol{\alpha}^{(t)}; \boldsymbol{\theta}^{(t)})$      `// estimated log likelihood`
6      $\boldsymbol{g}_c^{(t)} \leftarrow \texttt{AutoDiff}\left(-l^{(t)}, \boldsymbol{\theta}^{(t)}\right)$     `// gradient of NLL w.r.t. classifier weights`
7      $\boldsymbol{\theta}^{(t+1)} \leftarrow \texttt{OptimizerStep}(\boldsymbol{\theta}^{(t)}, \boldsymbol{g}_c^{(t)})$
8      $\boldsymbol{\gamma}^{(t)} \leftarrow \text{Sigmoid}(\boldsymbol{\gamma}'^{(t)})$        `// erasure probabilities`
9      $\boldsymbol{\alpha}^{(t+0.5)} \leftarrow \left(\alpha_\tau^{(t+0.5)} \sim \text{Bernoulli}(p = 1 - \gamma_\tau^{(t)}) : \tau = 1, \ldots, T\right)$     `// binary noise`
10      $l^{(t+0.5)} \leftarrow \log \Phi(y^{(n)} \mid \boldsymbol{\alpha}^{(t+0.5)} \odot \boldsymbol{x}^{(n)}, \boldsymbol{\alpha}^{(t+0.5)}; \boldsymbol{\theta}^{(t+1)})$     `// new estimated log likelihood`
11      $\boldsymbol{g}_o^{(t)} \leftarrow \lambda\boldsymbol{\gamma}^{(t)} \odot \boldsymbol{\gamma}^{(t)} \odot (\mathbf{1}-\boldsymbol{\gamma}^{(t)}) + l^{(t+0.5)}\left((\mathbf{1} - \boldsymbol{\alpha}^{(t+0.5)}) \odot (\mathbf{1} - \boldsymbol{\gamma}^{(t)}) - \boldsymbol{\alpha}^{(t+0.5)} \odot \boldsymbol{\gamma}^{(t)}\right)$     `// gradient of loss w.r.t. unsquashed erasure probabilities`
12      $\boldsymbol{\gamma}'^{(t+1)} \leftarrow \texttt{OptimizerStep}(\boldsymbol{\gamma}'^{(t)}, \boldsymbol{g}_o^{(t)})$
13      $t \leftarrow t + 1$
14   **return** $\boldsymbol{\theta}^{(t)}, \boldsymbol{\gamma}'^{(t)}$

---

2022; Perin et al., 2022; Schamberger et al., 2023; Yap et al., 2023; Li et al., 2024). As baselines we compare our method to Gradient Visualization (Masure et al., 2019), input $*$ gradient (Shrikumar et al., 2017) (applied to side-channel attacks by Wouters et al. (2020)), and input occlusion (Zeiler & Fergus, 2014) (applied to side-channel attacks by Hettwer et al. (2020)), as they are neural net architecture-agnostic and widely used in the side-channel attack literature. Whereas these approaches simply perform supervised deep learning in the conventional manner and then interpret their trained neural net's outputs, we train our classifier in an unconventional but principled manner so that it can be used to estimate conditional mutual information quantities. Compared to this prior work, we expect our approach to perform better when conventionally-trained neural nets learn shortcuts (Geirhos et al., 2020; Hermann & Lampinen, 2020) and fail to leverage all available key-trace associations.

It is common to use first-order statistical techniques to estimate quantities similar to $\mathbb{I}[Y; X_t]$. As baselines we consider correlation power analysis (Brier et al., 2004) and the sum of squared differences technique (Chari et al., 2003) because their leakage estimates have been found to correlate well with Gaussian template attack performance (Fan et al., 2014), as well as the signal-to-noise ratio technique (Mangard et al., 2007) due to its ubiquity in the power side-channel attack literature. In contrast to our method, these techniques are unable to exploit higher-order statistical associations and make strong assumptions about the form of $p_{\boldsymbol{X},Y}$. Higher-order statistical techniques exist, but still require strong assumptions, and tend to either have exponential runtime in the maximum-considered order of association, assume the existence of device flaws, or assume unrealistic knowledge of internal random variables or the points in time at which they are operated on (Messerges, 2000; Agrawal et al., 2005).

## 5 EXPERIMENTAL RESULTS

**Experiments on synthetic AES datasets**   We first apply our technique to synthetic AES power trace datasets generated using the Hamming weight leakage model of Mangard et al. (2007) (see figure 3). We verify that results are consistent with ground-truth leaking points which we know by virtue of having generated the datasets ourselves. This holds for an unprotected AES implementation, as well as when we simulate the common random delay (Coron & Kizhvatov, 2009), random shuffling

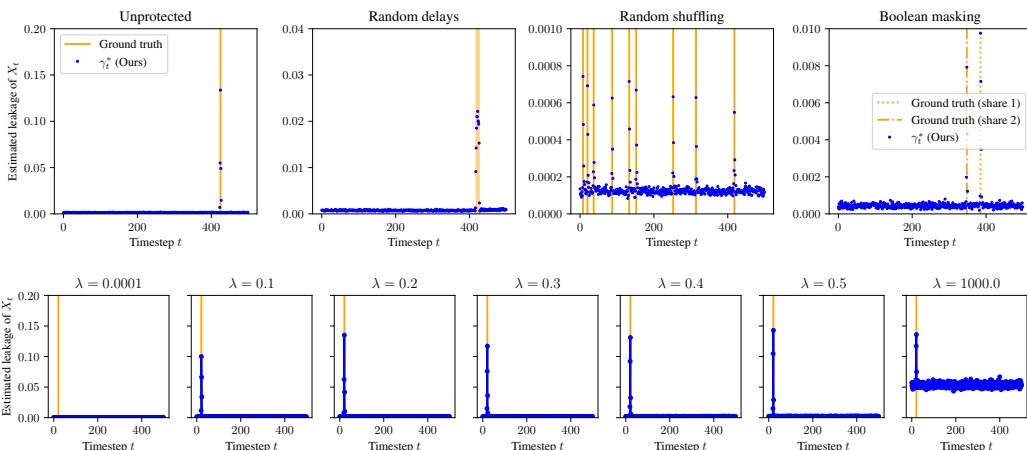

Figure 3: **(top row)** Adversarial leakage localization (our technique) is able to detect leakage from simulated AES implementations with various realistic countermeasures. Yellow lines denote ground-truth leaking points, and blue dots denote the estimated leakage $\lambda\gamma_t^*$ by our technique. **(bottom row)** As predicted by Proposition 1, for 'reasonable' values of $\lambda$, $\lambda\gamma_t^*$ is approximately constant as $\lambda$ is varied. Here we sweep $\lambda$ for a simulated unprotected AES implementation. Observe that results are nearly the same for $\lambda \in \{0.2, 0.3, 0.4, 0.5\}$. For $\lambda \in \{0.0001, 0.1\}$, the estimated leakage is smaller because $\gamma_t^*$ has saturated at 1. For $\lambda = 1000$, non-leaky points are more underfit than leaky points because all values of $\gamma_t^*$ lie far into the lower saturation region of Sigmoid .

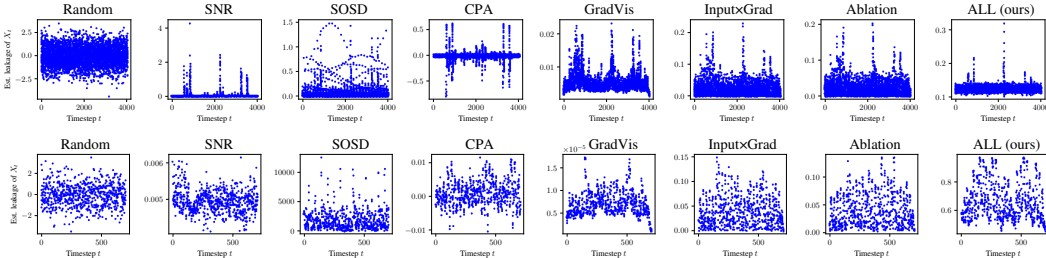

Figure 4: Qualitative comparison of our leakage localization technique with various baselines. Blue dots denote the estimated leakage assigned to each $X_t$ by the considered methods. **(top row)** DPAv4 dataset. **(bottom row)** ASCADv1 (fixed key) dataset. In the leftmost column, as a baseline we have a random assignment of leakage values. The following 3 columns show results for the following first-order statistical techniques: signal-noise ratio (SNR), sum of squared differences (SOSD), correlation power analysis with a Hamming weight leakage model (CPA). The next 3 columns show neural network interpretation-based techniques: gradient visualization (GradVis), input ∗ gradient, and input ablation. The rightmost column shows our adversarial leakage localization (ALL) technique.

(Masure & Strullu, 2023), and Boolean masking (Benadjila et al., 2020) countermeasures which are often present in actual AES implementations. Additionally, we sweep the hyperparameter $\lambda$ to verify the implication of Proposition 2 that our algorithm's output is inversely proportional to $\lambda$ (except for extreme values which lead to underfitting or saturation of erasure probabilities at 1).

**Experiments on recorded power traces from real AES implementations**  We next evaluate the performance of our algorithm on 2 publicly-available datasets of recorded AES power traces paired with plaintexts, ciphertexts, and cryptographic keys. We use the fixed-key variant of the ASCADv1 database (Benadjila et al., 2020) based on a Boolean-masked AES implementation, as well as the DPAv4 (Nassar et al., 2012) subset released by Zaid et al. (2020) which is modified to effectively be unprotected. These datasets were chosen because they have fairly-localized side channel leakage, whereas many other commonly-evaluated public datasets are cropped to contain mostly leaking points and are thus poor choices for evaluating a leakage localization algorithm.

| | Method | DPAv4 dataset | ASCADv1 dataset |
|---|---|---|---|
| | Random | $0.000330 \pm 0.00371$ | $-0.0186 \pm 0.0378$ |
| First-order statistics | Signal-noise ratio | $0.0489$ | $-0.0918$ / $\mathbf{0.198}^{\dagger}$ |
| | Sum of squared differences | $\mathbf{0.0740}$ | $-0.0105$ / $0.193^{\dagger}$ |
| | Correlation power analysis | $0.0586$ | $-0.0363$ / $0.188^{\dagger}$ |
| Neural net interpretation | Gradient Visualization | $0.0622 \pm 0.00410$ | $\mathbf{0.136 \pm 0.0162}$ |
| | Input $*$ gradient | $0.0507 \pm 0.00230$ | $0.0869 \pm 0.0139$ |
| | Input ablation | $0.0512 \pm 0.00216$ | $0.0580 \pm 0.0180$ |
| | Adversarial leakage localization (Ours) | $0.0401 \pm 0.00256$ | $\mathbf{0.138 \pm 0.00729}$ |

Table 1: Performance comparison (higher is better) on datasets of recorded power traces according to the metric proposed in appendix C.2.1. For stochastic techniques, results are reported as mean $\pm$ standard deviation over 5 repetitions of the trial. The first-order statistical techniques perform similarly to the random baseline on ASCADv1 because this dataset has only second-order leakage. Apart from this, all methods far outperform the random baseline on both datasets. Our method achieves the best performance on ASCADv1 and the worst on DPAv4. We emphasize, however, that our metric only accounts for low-order statistical associations which can be represented by a Gaussian mixture model, and unavoidably discards nuance and information by summarizing performance with a scalar. Thus, while the reported numbers correlate with the fidelity of a leakage localization attempt, they should not be viewed as an oracle for fidelity. $^{\dagger}$Ground truth-like performance when we effectively disable the Boolean masking countermeasure using unrealistic knowledge of internal random numbers generated by the AES hardware.

Unlike for our synthetic datasets, here we lack ground truth knowledge about which power measurements are leaking, and it is not established or obvious how to evaluate the fidelity of an attempt to localize leakage. Masure et al. (2019) and Hettwer et al. (2020) propose several metrics and heuristics based on the intuition that the 'leakiness' assigned to a set of measurements should positively-correlate with the performance of a side-channel attack carried out using only those measurements. We propose our own metric (see appendix C.2.1) which is inspired by these but addresses shortcomings that they have. The rough idea is to perform many Gaussian template attacks (Chari et al., 2003) on subsets of available power measurements and compute the expected Kendall $\tau$ rank correlation coefficient (Kendall, 1938) between a set of points and its estimated leakage. A higher expected rank correlation indicates that estimated-leakier points are indeed more exploitable to side-channel attackers, and thus indicates higher fidelity of the leakage localization attempt.

In table 1 we report the performance of our method and considered baselines. Figure 4 contains plots of the amount of leakage at each timestep as estimated by each method. We find qualitatively-similar results for all methods apart from our random baseline, apart from the first-order statistical methods on ASCADv1 which are unable to detect the second-order statistical associations which result from Boolean masking. It appears that the deep learning-based methods assign significant leakage to non-leaky timesteps, likely due to overfitting. Our synthetic dataset results suggest that this issue may be alleviated in settings where we have access to infinitely-large datasets (e.g. in the pre-manufacturing phase where we can simulate arbitrarily-many synthetic power traces for a design).

## 6 CONCLUSION

We have proposed a novel, principled algorithm for learning to localize power side channel leakage from cryptographic algorithms, and have demonstrated its efficacy on real and synthetic implementations of the ubiquitous AES cryptographic standard. Our algorithm is generic enough to be applicable to other cryptographic standards as well. The growing assortment of deep learning-based side-channel attacks departs from classical attacks in that the DL-based attacks can largely treat cryptographic hardware as a black box to be characterized using data, whereas classical techniques required strong assumptions and knowledge about the particular device to be attacked. However, despite the demonstrated ability of deep learning to exploit side-channel leakage without *a priori* knowledge about its existence or nature, little work thus far exists on leveraging this ability to understand *why* devices leak, which is critical for designing countermeasures against side-channel attacks. Our work represents an initial step towards using deep learning to uncover power side-channel leakage without relying on human understanding of the complicated and non-ideal device physics by which the leakage happens.

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

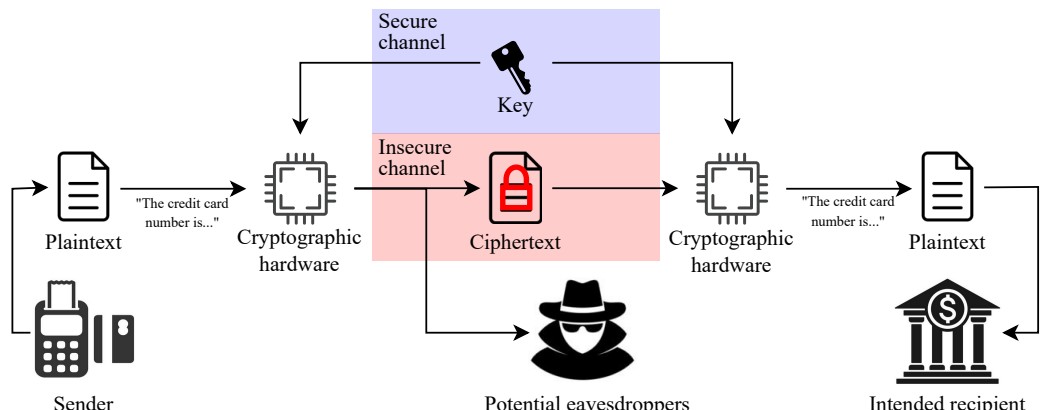

Figure 5: Diagram illustrating the main components of symmetric-key cryptographic algorithms, which enable secure transmission of data over insecure channels where it may be intercepted by eavesdroppers. The data is first partitioned and encoded as a sequence of plaintexts. Each plaintext is transformed into a ciphertext by an invertible function indexed by a cryptographic key. The key is transmitted over a secure channel to intended recipients of the data, allowing them to invert the function and recover the original plaintext. The set of functions is designed so that absent this key, the ciphertext gives no information about the plaintext. Thus, the data remains secure even if eavesdroppers have access to the ciphertext.

## A  EXTENDED BACKGROUND

Here we provide a high-level overview of the AES algorithm and power side-channel attacks aimed at a machine learning audience. Since our algorithm views the cryptographic algorithm and hardware as a black box to be characterized with data, a deep understanding is not necessary to understand and appreciate our work. Thus, we omit many details and aim to impart an intuitive understanding of these topics. Interested readers may refer to Daemen & Rijmen (2013) for a detailed introduction to the AES algorithm, to Mangard et al. (2007) for a detailed introduction to power side-channel attacks, and to Picek et al. (2023) for a survey of supervised deep learning-based power side-channel attacks on AES implementations.

### A.1  CRYPTOGRAPHIC ALGORITHMS

Data is often transmitted over insecure channels which leave it accessible not only to intended recipients, but also to unknown and untrusted parties. For example, when a signal is wirelessly transmitted from one antenna to another, an eavesdropper could set up a third antenna between the two and intercept the signal. Alternately, data stored on a hard drive by one user of a computer may be accessed by a different user. Cryptographic algorithms aim to preserve the privacy of data under such circumstances by transforming it so that it is meaningful only in combination with additional data which is known to its intended recipients but not to the untrusted parties.

In this work we consider the advanced encryption standard (AES), which is a symmetric-key cryptographic algorithm. See figure 5 for a diagram illustrating the important components of such algorithms. The unencrypted data to be transmitted is encoded and partitioned into a sequence of fixed-length bitstrings called *plaintexts*. The cryptographic algorithm encrypts each plaintext into a *ciphertext* by applying an invertible function from a set of functions indexed by an integer called the *cryptographic key*. This set of functions is designed so that of one were to sample a key and plaintext uniformly at random from the sets of all possible keys and plaintexts, then the plaintext and ciphertext would be marginally independent. Thus, such an algorithm may be used to securely transmit data by ensuring that the sender and recipient of the data know a shared key,[2] and that the key is kept secret from all potential eavesdroppers on the data.

---

[2]The key is typically shared using an asymmetric-key cryptographic algorithm such as RSA or ECC. Asymmetric-key cryptography is slow and resource-intensive, so when a sufficiently-large amount of data must

Many symmetric-key cryptographic algorithms are believed to be secure in the sense that it is not feasible to determine their cryptographic key by encrypting known plaintexts and observing the resulting ciphertexts. Any such algorithm with a finite number of possible keys is vulnerable to 'brute-force' attacks based on arbitrarily guessing and checking keys until success, but doing so requires checking half of all possible keys in the average case, which is unrealistic for algorithms such as AES which has either $2^{128}$, $2^{192}$, or $2^{256}$ possible keys. To our knowledge the best known such attack against AES reduces the required number of guesses by less than a factor of $8$ compared to a naive brute force attack (Mouha, 2021; Tao & Wu, 2015).

However, while algorithms may be secure when considering only their intended inputs and outputs, *hardware executing these algorithms* will inevitably emit measurable physical signals which are statistically associated with their intermediate variables and operations. Examples of such signals include a device's power consumption over time (Kocher et al., 1999), the amount of time it takes to execute a program or instruction (Kocher, 1996; Lipp et al., 2018; Kocher et al., 2019), electromagnetic radiation it emits (Quisquater & Samyde, 2001; Genkin et al., 2016), and sound due to vibrations of its electronic components (Genkin et al., 2014). This phenomenon is called *side-channel leakage*, and can be exploited to determine sensitive data such as a cryptographic key through *side-channel attacks*.

As a simple example of side-channel leakage, consider the following Python function which checks whether a password is correct:

```python
def is_correct(provided_password: str, correct_password: str) -> bool:
    if len(provided_password) != len(correct_password):
        return False
    for i in range(len(provided_password)):
        if provided_password[i] != correct_password[i]:
            return False
    return True
```

Suppose the password consists of $n$ characters, each with $c$ possible values. Consider an attacker seeking to determine the correct password by feeding various guessed passwords until the function returns `True`. Naively, the attacker could simply guess and check all possible $m$-length passwords for $m = 1, \ldots, n$. This would require $\mathcal{O}(c^n)$ calls to the function, which would be extremely costly for realistically-large $c$ and $n$. However, an attacker with knowledge of the function's implementation could dramatically reduce this cost by observing that the function's *execution time* depends on `correct_password`. Because the function exits immediately if `len(provided_password) != len(correct_password)`, the attacker can determine the length of `correct_password` in $\mathcal{O}(n)$ time by feeding increasing-length guesses to `is_correct` until its execution time increases. Next, because `is_correct` exits the first time it detects an incorrect character, the attacker can sequentially determine each of the characters of `correct_password` by checking all $c$ possible values of each character and noting that the correct value leads to an increase in execution time. Thus, although `is_correct` secure against attackers which use only its intended inputs and outputs, it provides *essentially no security* against attackers which measure its execution time.

In this work we focus on side-channel leakage due to the power consumption over time of a device. A device's power consumption is inevitably statistically-associated with the operations it executes and the data it operates on, because these dictate which components are active and the order and manner in which they operate. There are many types of components with different functionality, and components with the same intended functionality are not identical due to imperfect manufacturing processes. These differences impact power consumption. While in general the association between power consumption and data is multifactorial and difficult to describe, in figure 6 we illustrate a simple relationship which accounts for a significant portion of the leakage in a device characterized by Mangard et al. (2007).

---

be transmitted, it is more-efficient to share the key with an asymmetric-key algorithm and then transmit data using a symmetric-key algorithm than to simply transmit the data with an asymmetric-key algorithm.

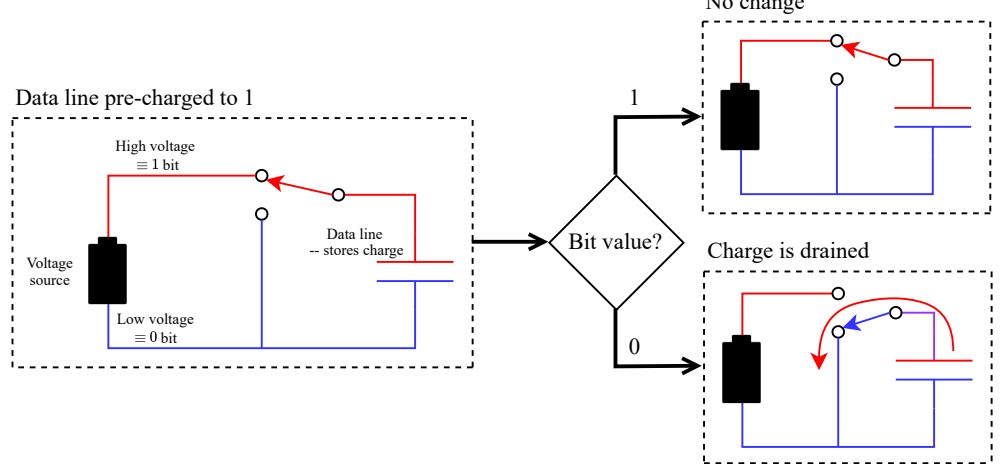

Figure 6: Diagram illustrating one reason there is power side-channel leakage in the device characterized by Mangard et al. (2007, ch. 4). Data is transmitted over a bus consisting of multiple wires, with one wire representing each bit. Each wire represents a 0 bit as some prescribed 'low' voltage and a 1 bit as a 'high' voltage. Energy is consumed when the voltage of a wire changes from low to high because positive and negative charges, which are attracted to one-another, must be separated to create a high concentration of positive charge on the wire. When 'writing' data to the bus, this particular device first 'pre-charges' all wires to 1, then drains charge from the wires which should represent 0. Thus, because the 0's must be changed to 1's before the next write, energy is consumed in proportion to the number of 0's, thereby creating a statistical association between the device's power consumption and the data it operates on.

## A.3   POWER SIDE-CHANNEL ATTACKS ON AES IMPLEMENTATIONS

*Side-channel attacks* are techniques which exploit side-channel leakage to learn sensitive information such as cryptographic keys. There are many categories of attacks, but in this work we focus on a category called *profiled* side-channel attacks on symmetric-key cryptographic algorithms. These attacks assume that the 'attacker' has access to a clone of the actual cryptographic device to be attacked, and the ability to encrypt arbitrary plaintexts with arbitrary cryptographic keys, observe the resulting ciphertexts, and measure the side-channel leakage during encryption. In practice, these assumptions almost certainly overestimate the capabilities of attackers – for example, while in some cases an attacker could plausibly identify the hardware and source code of a cryptographic implementation, purchase copies of this hardware, program them with the source code, and characterize these devices, the nature of the side-channel leakage of these purchased copies would differ from those of the actual device due to imperfect manufacturing processes. It has been demonstrated that profiled side-channel attacks can be effective despite this, especially when numerous copies of the target hardware are used for profiling (Das et al., 2019; Danial et al., 2021). Regardless, this type of attack provides an upper bound on the vulnerability of a device to side-channel attacks, which is a useful metric for hardware designers.

While there are diverse types of profiled side-channel attacks, at a high level the following steps encompass the important elements of these attacks:

1. Select some 'sensitive' intermediate variable of the cryptographic algorithm which reveals the cryptographic key (or part of it).

2. Compile a dataset of (side-channel leakage, intermediate variable) pairs by repeatedly randomly selecting a key and plaintext, encrypting the plaintext using the key and recording the resulting ciphertext and side-channel leakage during encryption, and computing the intermediate variable based on knowledge of the cryptographic algorithm.

3. Use supervised learning to train a parametric function approximator to predict intermediate variables from recordings of side-channel leakage during encryption.

4. Measure side-channel leakage during encryptions by the actual target device. Use the trained predictor to predict sensitive variables from side-channel leakage. Potentially, these predictions can be combined to get a better estimate of the key.

In the case of power side-channel attacks on AES, it is generally infeasible to directly target the cryptographic key because care is taken by hardware designers to prevent it from directly influencing power consumption. Instead, it is common to target an intermediate variable called the SubBytes output, which is computed as

$$y := \text{AES-SBOX}(k \oplus w) \tag{15}$$

where $k \in \{0,1\}^{n_{\text{bits}}}$ is the key, $w \in \{0,1\}^{n_{\text{bits}}}$ is the plaintext, $n_{\text{bits}} \in \mathbb{Z}_{++}$ is the number of bits of the key and plaintext, $\oplus$ is the bitwise exclusive-or operation, and $\text{AES-SBOX} : \{0,1\}^{n_{\text{bits}}} \to \{0,1\}^{n_{\text{bits}}}$ is an invertible function which is widely known and the same for all AES implementations. Note that if the plaintext is known, the key can be computed as

$$k = \text{AES-SBOX}^{-1}(y) \oplus w. \tag{16}$$

Additionally, it is common to independently target subsets of the bits of the cryptographic key (e.g. the individual bytes). This is reasonable because many devices can operate on only a subset of the bytes in a single machine instruction, in which case one gains little by attacking more than this number of bytes simultaneously. Even in devices for which this is not the case, subsets of bits will still be statistically associated with power consumption.

### A.3.1 TEMPLATE ATTACK: EXAMPLE OF A CLASSICAL PROFILED SIDE-CHANNEL ATTACK

In order to underscore the advantage of deep learning over previous side-channel attack algorithms, we will here describe the template attack algorithm of Chari et al. (2003), variations of which are the state-of-the-art non-deep learning based attacks. The attack is based on modeling the joint distribution of power consumption and intermediate variable as a Gaussian mixture model, as described in algorithm 2.

---

**Algorithm 2:** The Gaussian template attack algorithm of Chari et al. (2003)

---

**Input:** Profiling (training) dataset $\mathsf{D} := \{(\boldsymbol{x}^{(n)}, y^{(n)} : n \in [1 .. N]\} \subset \mathbb{R}^T \times \{0,1\}^{n_{\text{bits}}}$, attack (testing) dataset $\mathsf{D}_{\text{attack}} := \{(\boldsymbol{x}_{\text{a}}^{(n)}, w_{\text{a}}^{(n)}) : n \in [1 .. N_{\text{a}}]\} \subset \mathbb{R}^T \times \{0,1\}^{n_{\text{bits}}}$, 'points of interest' $\boldsymbol{T}_{\text{poi}} := \{t_m : m = 1, \ldots, \tilde{T}\} \subset [1 .. T]$

**Output:** Predicted key $k^*$

1 **Function** get_y *(k, w)*
2     **return** $\text{AES-SBOX}(k \oplus w)$    // calculate intermediate variable for given key
3 **for** $n \in [1 .. N]$ **do**
4     $\tilde{\boldsymbol{x}}^{(n)} \leftarrow \left( x_{t_m}^{(n)} : m = 1, \ldots, \tilde{T} \right)$    // prune power traces to 'points of interest'
5 **for** $y \in \{0,1\}^{n_{\text{bits}}}$ **do**
6     // fit a multivariate Gaussian mixture model to the training dataset
    $\mathsf{D}_y \leftarrow \left\{ \tilde{\boldsymbol{x}}^{(n)} : n \in [1 .. N], y^{(n)} = y \right\}$
7     $N_y \leftarrow |\mathsf{D}_y|$
8     $\boldsymbol{\mu}_y \leftarrow \frac{1}{N_y} \sum_{\tilde{\boldsymbol{x}} \in \mathsf{D}_y} \tilde{\boldsymbol{x}}$
9     $\boldsymbol{\Sigma}_y \leftarrow \frac{1}{N_y - 1} \sum_{\tilde{\boldsymbol{x}} \in \mathsf{D}_y} (\tilde{\boldsymbol{x}} - \boldsymbol{\mu}_y)(\tilde{\boldsymbol{x}} - \boldsymbol{\mu}_y)^\top$
10 **for** $n \in [1 .. N_a]$ **do**
11     $\tilde{\boldsymbol{x}}_{\text{a}}^{(n)} \leftarrow \left( x_{\text{a}, t_m}^{(n)} : m = 1, \ldots, \tilde{T} \right)$      // prune power traces of attack dataset

    // predict key value which maximizes log-likelihood of attack dataset
12 $k^* \leftarrow$
    $\arg\max_{k \in \{0,1\}^{n_{\text{bits}}}} \sum_{n=1}^{N_{\text{a}}} \left[ \log \mathcal{N} \left( \tilde{\boldsymbol{x}}^{(n)}; \boldsymbol{\mu}_{\text{get\_y}(k, w_{\text{a}}^{(n)})}, \boldsymbol{\Sigma}_{\text{get\_y}(k, w_{\text{a}}^{(n)})} \right) + \log N_{\text{get\_y}(k, w_{\text{a}}^{(n)})} \right]$
13 **return** $k^*$

---

Note that this algorithm assumes that the joint distribution is well-described by a Gaussian mixture model, which may not hold in practice. Additionally, due to the near-cubic runtime of the matrix inversion of each $\boldsymbol{\Sigma}_y$ required to compute the Gaussian density functions, this algorithm requires pruning power traces down to a small number of 'high-leakage' timesteps. Follow-up work (Rechberger & Oswald, 2005) proposed performing principle component analysis on the traces and modeling the coefficients of the top principle components rather than individual timesteps. Nonetheless, these constraints mean that the efficacy of this attack is contingent on simplifying assumptions and judgement of which points are 'leaky' using simple statistical techniques and implementation knowledge, limiting its usefulness as a way for hardware designers to evaluate the amount of side-channel leakage from their device.

### A.3.2  PRACTICAL PROFILED DEEP LEARNING SIDE-CHANNEL ATTACKS ON AES IMPLEMENTATIONS

Here we will give a common and concrete setting and method for performing profiled power side-channel attacks on AES implementations, which is used for all of our experiments.

Consider an AES-128 implementation, which has a 128-bit cryptographic key and plaintext. Typically, attackers target each of the 16 bytes of the key independently rather than attacking the full key at once. This practice tacitly assumes that the bytes of the sensitive variable are statistically-independent given the power trace, which is reasonable because many AES operations (including those which are commonly targeted) are performed independently on the individual bytes. Thus, it is a convenient way to simplify the attack with only a small performance degradation.

Additionally, it is difficult and uncommon to try to directly map power traces to associated cryptographic keys, because great care is taken by hardware designers to ensure that the key does not directly impact power consumption. Instead, attackers generally target 'sensitive' intermediate variables which unavoidably directly impact power consumption and can be combined with the plaintext and ciphertext to learn the key. We consider one such intermediate variable which is referred to as the first SubBytes output, and is equal to

$$y := \text{AES-SBOX}(k \oplus w), \tag{17}$$

where $k \in \{0,1\}^8$ is one byte of the cryptographic key, $w \in \{0,1\}^8$ is the corresponding byte of the plaintext, $\oplus$ denotes the bitwise exclusive-or operation, and $\text{AES-SBOX} : \{0,1\}^8 \to \{0,1\}^8$ is an invertible function which is publicly-available and the same for all AES implementations. Note that if $w$ is known, as is assumed in the profiled side-channel attack setting, then $k$ can be recovered as

$$k = w \oplus \text{AES-SBOX}^{-1}(y). \tag{18}$$

In the context of profiled power side-channel analysis, one assumes to have a 'profiling' dataset (i.e. a training dataset) and an 'attack' dataset (i.e. a test dataset). Suppose we target $n_{\text{bytes}}$ bytes of the sensitive variable. In our setting, the profiling dataset consists of ordered pairs of power traces and their associated sensitive intermediate variables:

$$\mathsf{D} := \left\{ (\boldsymbol{x}^{(n)}, y^{(n)}) : n \in [1 \mathbin{..} N] \right\} \subset \mathbb{R}^T \times \{0,1\}^{n_{\text{bytes}} \times 8} \tag{19}$$

and the attack dataset consists of ordered pairs of power traces and their associated plaintexts:

$$\mathsf{D}_{\text{a}} := \left\{ (\boldsymbol{x}_{\text{a}}^{(n)}, w_{\text{a}}^{(n)}) : n \in [1 \mathbin{..} N_{\text{a}}] \right\} \subset \mathbb{R}^T \times \{0,1\}^{n_{\text{bytes}} \times 8}. \tag{20}$$

Many works prove the concept of their approaches by targeting only a single byte of the sensitive variable. When multiple bytes are targeted, it is common to either train a separate neural network for each byte of the sensitive variable, or to amortize the cost of targeting these bytes by training a single neural network with a shared backbone and a separate head for each byte. In this work we exclusively target single bytes, though it would be straightforward to extend our approach to the multitask learning setting.

Consider a neural network architecture $\Phi : \mathsf{Y} \times \mathbb{R}^T \times \mathbb{R}^P \to \mathbb{R}_+ : (y, \boldsymbol{x}, \boldsymbol{\theta}) \mapsto \Phi(y \mid \boldsymbol{x}; \boldsymbol{\theta})$, where each $\Phi(\cdot \mid \boldsymbol{x}; \boldsymbol{\theta})$ is a probability mass function over $\mathsf{Y}$. In the case of a multi-headed network with each head independently predicting a single byte, we compute this probability mass of $y \in \mathsf{Y}$ as the

product of the mass assigned to each of its bytes. We train the network by approximately solving the optimization problem

$$\max_{\boldsymbol{\theta} \in \mathbb{R}^P} \quad \mathcal{L}(\boldsymbol{\theta}) := \frac{1}{N} \sum_{n=1}^{N} \log \Phi(y^{(n)} \mid \boldsymbol{x}^{(n)}; \boldsymbol{\theta}). \tag{21}$$

Given $\hat{\boldsymbol{\theta}} \in \arg\max_{\boldsymbol{\theta} \in \mathbb{R}^P} \mathcal{L}(\boldsymbol{\theta})$, we then identify the key which maximizes our estimated likelihood of our attack dataset and key as follows:

$$\hat{k} \in \arg\max_{k \in \{0,1\}^{n_{\text{bytes}} \times 8}} \sum_{n=1}^{N_{\text{a}}} \log \Phi \left( \left( \text{AES-SBOX}(k_i \oplus w_{\text{a},i}^{(n)}) : i = 1, \dots, n_{\text{bytes}} \right) \mid \boldsymbol{x}_{\text{a}}^{(n)}; \hat{\boldsymbol{\theta}} \right) \tag{22}$$

where we denote by $k_i$ and $w_i^{(n)}$ the individual bytes of $k$ and $w^{(n)}$.

**Model evaluation** In the context of profiled power side-channel analysis, the accuracy of trained models is usually only marginally higher than that of randomly guessing, and higher accuracy is achieved by accumulating predictions about many power traces in the manner of equation 22. Thus, accuracy generally lacks the resolution to usefully evaluate and compare models. Instead, it is common to estimate the rank of the correct key in the distribution predicted by the model, defined as $\mathbb{E} \operatorname{rank}(\Phi(\cdot \mid \boldsymbol{X}; \boldsymbol{\theta}), Y)$ where

$$\operatorname{rank}(f, y) := |\{y' \in \mathsf{Y} : f(y') \geq f(y)\}| - 1 \tag{23}$$

for $f : \mathsf{Y} \to \mathbb{R}$. This quantity is not equivalent to the performance of the model when accumulating predictions on multiple power traces, as performance in this regime may vary significantly depending on the extent to which the model's incorrect predictions are systematic or random. For multi-trace predictions, given the attack dataset $\mathsf{D}_{\text{a}}$ of equation 20 and arbitrary $\hat{\mathsf{D}}_{\text{a}} \subset \mathsf{D}_{\text{a}}$, let us define

$$\tilde{p}_K(k; \hat{\mathsf{D}}_{\text{a}}, \boldsymbol{\theta}) := \sum_{(\boldsymbol{x}, w) \in \hat{\mathsf{D}}_{\text{a}}} \log \Phi \left( (\text{AES-SBOX}(k_i \oplus w_i) : i = 1, \dots, n_{\text{bytes}}) \mid \boldsymbol{x}; \boldsymbol{\theta} \right), \tag{24}$$

i.e. a quantity proportional to the logarithm of the estimated distribution of the key by our model given our attack dataset. We define

$$\operatorname{rank-auc}(\boldsymbol{\theta}; t, \mathsf{D}_{\text{a}}, k) := \sum_{\tau=1}^{t} \mathbb{E}_{\hat{\mathsf{D}}_{\text{a}} \sim \mathcal{U}\binom{\mathsf{D}_{\text{a}}}{t}} \operatorname{rank}(\tilde{p}_K(\cdot; \hat{\mathsf{D}}_{\text{a}}, \boldsymbol{\theta}), k). \tag{25}$$

Intuitively, this tells us the area under the curve we would get if we were to evaluate our model's key predictions using random $\tau$-cardinality subsets of our attack dataset and compute the mean rank of key $k$ in these predictions, and plot this quantity for each $\tau \in [1 .. t]$. Lower values of this quantity when $k$ denotes the correct key tell us that our model has better performance in the regime where it is fed multiple power traces.

# B EXTENDED METHOD WITH PROOFS AND DERIVATIONS

## B.1 NOTATION

We denote sets with Serif font, e.g. $\mathsf{S}$, with the exception of the real numbers $\mathbb{R}$ and the integers $\mathbb{Z}$. For arbitrary sets $\mathsf{S} \subset \mathbb{R}$, we will define $\mathsf{S}_+ := \{x \in \mathsf{S} : x \geq 0\}$ and $\mathsf{S}_{++} := \{x \in \mathsf{S} : x > 0\}$. For $a \leq b \in \mathbb{Z}$, we will define $[a .. b] := [a, b] \cap \mathbb{Z}$. We will use set-builder notation when we wish to assign names to a set's elements, e.g. $\mathsf{S} := \{x_t : t \in [1 .. T]\}$.

We denote vectors with boldface text, e.g. $\boldsymbol{x}$, and scalars with non-bold text, e.g. $x$. We will denote by $x_t$ the element of vector $\boldsymbol{x}$ at position $t$. We will sometimes use the following 'vector-builder' notation to define elements of the vectors: $\boldsymbol{x} := (x_t : t = 1, \dots, T)$. We will denote by $\boldsymbol{x}^\top$ the transpose of $\boldsymbol{x}$. Note that in this work, $(\cdot)^T$ does not denote transposition, but rather some object to the power of another object $T$.

Random variables will always be upper-case, whereas deterministic variables may be either lower- or upper-case. In this work we will assume that all real-valued random variables have probability

density functions, and will denote by 'distribution' a probability mass, density, or joint mass/density function, depending on the context. Distributions will be denoted by $p_{...}$ with subscript indicating the nature of the distribution. For example, we may denote by $p_{\boldsymbol{X}}$ the distribution of random vector $\boldsymbol{X}$, by $p_{\boldsymbol{X},Y}$ the joint distribution of random vector $\boldsymbol{X}$ and scalar $Y$, or $p_{Y|\boldsymbol{X}}$ the conditional distribution of $Y$ given $\boldsymbol{X}$. We will denote expectation by $\mathbb{E}$.

For arbitrary vectors $\boldsymbol{x} \in \mathbb{R}^T$ and binary vectors $\boldsymbol{\alpha} \in \{0,1\}^T$, we denote $\boldsymbol{x}_{\boldsymbol{\alpha}} := (x_t : t = 1, \ldots, T : \alpha_t = 1)$, i.e. the sub-vector of $\boldsymbol{x}$ containing its elements for which the corresponding element of $\boldsymbol{\alpha}$ is 1. We will denote by $\boldsymbol{x}_{-t} := (x_\tau : \tau = 1, \ldots, T : \tau \neq t)$, i.e. the vector $\boldsymbol{x}$ with element $t$ omitted.

We will use the following Shannon information theoretic quantities (Shannon, 1948):

$$
\begin{aligned}
\mathbb{H}[X] &:= \mathbb{E} \log p_X(X) \quad \text{for discrete } X, & \text{(entropy)} \\
\mathbb{I}[X;Y] &:= \mathbb{E} \left[ \log p_{X,Y}(X,Y) - \log p_X(X) - \log p_Y(Y) \right]. & \text{(mutual information)} \\
\mathbb{KL}[p \,\|\, q] &:= \mathbb{E}_{X \sim p} \left[ \log p(X) - \log q(X) \right] & \text{(KL divergence)}
\end{aligned}
$$

Conditional entropies are defined similarly: $\mathbb{H}[Y \mid X] := \mathbb{E} \log p_{Y|X}(Y \mid X)$. Note that $\mathbb{I}[X;Y] = \mathbb{H}[X] - \mathbb{H}[X \mid Y] = \mathbb{H}[Y] - \mathbb{H}[Y \mid X]$. While we have used random scalar notation in these definitions, they are equally-applicable to random vectors.

## B.2 Setting

We view power traces as vectors $\boldsymbol{x} \in \mathbb{R}^T$ and sensitive variable values as elements $y \in \mathsf{Y}$ where $\mathsf{Y}$ is a finite set. We view these as realizations of jointly-distributed random variables $\boldsymbol{X}, Y \sim p_{\boldsymbol{X},Y}$ respectively, where $p_Y$ is a simple known (e.g. uniform) distribution and $p_{\boldsymbol{X}|Y}$ is *a priori* unknown and dictated by factors such as the hardware, environment, and measurement setup. In this work we assume that the conditional density functions $p_{\boldsymbol{X}|Y}(\cdot \mid y)$ exist and have support equal to $\mathbb{R}^T$, which is reasonable because power consumption usually has a 'random' component which is well-described by additive Gaussian noise (Mangard et al., 2007).

## B.3 Optimization problem

Recall that we have defined the erasure probabilities to be a binary vector $\boldsymbol{\gamma} \in [0,1]^T$. This vector parameterizes a distribution over binary vectors in $\{0,1\}^T$ as follows:

$$
\mathcal{A}_{\boldsymbol{\gamma}} \sim p_{\mathcal{A}_{\boldsymbol{\gamma}}} \quad \text{where} \quad \mathcal{A}_{\boldsymbol{\gamma},t} = \begin{cases} 1 & \text{with probability} \quad 1 - \gamma_t \\ 0 & \text{with probability} \quad \gamma_t \end{cases}, \tag{26}
$$

i.e. its elements are independent Bernoulli random variables where the $t$-th element has parameter $p = 1 - \gamma_t$. We will use $\mathcal{A}_{\boldsymbol{\gamma}}$ to get random sub-vectors $\boldsymbol{X}_{\mathcal{A}_{\boldsymbol{\gamma}}}$ of $\boldsymbol{X}$. Note that $\gamma_t$ represents the probability that $X_t$ will *not* be an element of $\boldsymbol{X}_{\mathcal{A}_{\boldsymbol{\gamma}}}$.

We consider the optimization problem

$$
\min_{\boldsymbol{\gamma} \in [0,1]^T} \quad \mathcal{L}_{\text{ideal}}(\boldsymbol{\gamma}) := \frac{1}{2} \lambda \|\boldsymbol{\gamma}\|_2^2 + \mathbb{I}[Y; \boldsymbol{X}_{\mathcal{A}_{\boldsymbol{\gamma}}} \mid \mathcal{A}_{\boldsymbol{\gamma}}] \tag{27}
$$

where $\lambda > 0$ is a hyperparameter. Intuitively, this is a trade-off between having high erasure probabilities and thereby reducing the mutual information quantity, and having low erasure probabilities to

reduce the norm penalty. For each $t$, we can write

$$\mathcal{L}_{\text{ideal}}(\boldsymbol{\gamma}) = \frac{1}{2}\lambda \sum_{\tau=1}^{T} \gamma_\tau^2 + \sum_{\boldsymbol{\alpha} \in \{0,1\}^T} \mathbb{I}[Y; \boldsymbol{X}_{\boldsymbol{\alpha}}] p_{\mathcal{A}_{\boldsymbol{\gamma}}}(\boldsymbol{\alpha}) \tag{28}$$

$$= \frac{1}{2}\lambda \sum_{\tau=1}^{T} \gamma_\tau^2 + \sum_{\substack{\boldsymbol{\alpha} \in \{0,1\}^T \\ \alpha_t=1}} p_{\mathcal{A}_{\boldsymbol{\gamma},t}}(1) p_{\mathcal{A}_{\boldsymbol{\gamma}}}(\boldsymbol{\alpha}_{-t}) \, \mathbb{I}[Y; \boldsymbol{X}_{\boldsymbol{\alpha}}]$$

$$+ \sum_{\substack{\boldsymbol{\alpha} \in \{0,1\}^T \\ \alpha_t=0}} p_{\mathcal{A}_{\boldsymbol{\gamma},t}}(0) p_{\mathcal{A}_{\boldsymbol{\gamma},-t}}(\boldsymbol{\alpha}_{-t}) \, \mathbb{I}[Y; \boldsymbol{X}_{\boldsymbol{\alpha}}] \tag{29}$$

$$= \frac{1}{2}\lambda \sum_{\tau=1}^{T} \gamma_\tau^2 + \sum_{\substack{\boldsymbol{\alpha} \in \{0,1\}^T \\ \alpha_t=0}} \left[ (1-\gamma_t) \, \mathbb{I}[Y; \boldsymbol{X}_{\boldsymbol{\alpha}}, X_t]] + \gamma_t \, \mathbb{I}[Y; \boldsymbol{X}_{\boldsymbol{\alpha}}]\right] p_{\mathcal{A}_{\boldsymbol{\gamma},-t}}(\boldsymbol{\alpha}_{-t}) \tag{30}$$

$$= \frac{1}{2}\lambda \gamma_t^2 + \sum_{\substack{\boldsymbol{\alpha} \in \{0,1\}^T \\ \alpha_t=0}} \left[ \mathbb{I}[Y; X_t, \boldsymbol{X}_{\boldsymbol{\alpha}}] - \gamma_t \, \mathbb{I}[Y; X_t \mid \boldsymbol{X}_{\boldsymbol{\alpha}}]\right] p_{\mathcal{A}_{\boldsymbol{\gamma},-t}}(\boldsymbol{\alpha}_{-t}) + \frac{1}{2}\lambda \left\| \boldsymbol{\gamma}_{-t} \right\|_2^2 . \tag{31}$$

As $\gamma_t$ decreases, so does the penalty $\frac{1}{2}\lambda \gamma_t^2$. As $\gamma_t$ increases, the quantities $-\gamma_t \, \mathbb{I}[Y; X_t \mid \boldsymbol{X}_{\boldsymbol{\alpha}}]$ decrease in proportion to $\mathbb{I}[Y; X_t \mid \boldsymbol{X}_{\boldsymbol{\alpha}}]$. For $\boldsymbol{\gamma}^* \in \arg\min_{\boldsymbol{\gamma} \in [0,1]^T} \mathcal{L}_{\text{ideal}}(\boldsymbol{\gamma})$, it follows that $\gamma_t^*$ positively correlates with some notion of a 'typical' value of $\mathbb{I}[Y; X_t \mid \boldsymbol{X}_{\boldsymbol{\alpha}}]$, and it appears reasonable to view it as a notion of the 'amount of leakage' at time $t$.

We can verify that a solution to equation 27 exists, and derive an implicit expression for it:

**Proposition 1.** *For $\mathcal{L}_{ideal}$ as defined in equation 27, $\arg\min_{\boldsymbol{\gamma} \in [0,1]^T} \mathcal{L}_{ideal}(\boldsymbol{\gamma}) \neq \emptyset$. Furthermore, every $\boldsymbol{\gamma}^* \in \arg\min_{\boldsymbol{\gamma} \in [0,1]^T} \mathcal{L}_{ideal}(\boldsymbol{\gamma})$ must satisfy*

$$\gamma_t^* = \min \left\{ \frac{1}{\lambda} \sum_{\substack{\boldsymbol{\alpha} \in \{0,1\}^T \\ \alpha_t=0}} \mathbb{I}[Y; X_t \mid \boldsymbol{X}_{\boldsymbol{\alpha}}] \prod_{\tau \in [1\,..\,T] \setminus \{t\}} (\gamma_\tau^*)^{1-\alpha_\tau} (1-\gamma_\tau^*)^{\alpha_\tau}, \quad 1 \right\} \qquad \forall t \in [1\,..\,T]. \tag{32}$$

*Proof.* To establish existence of a solution, we observe that $[0,1]^T$ is compact and

$$\mathbb{I}[Y; \boldsymbol{X}_{\mathcal{A}_{\boldsymbol{\gamma}}} \mid \mathcal{A}_{\boldsymbol{\gamma}}] = \sum_{\boldsymbol{\alpha} \in \{0,1\}^T} \mathbb{I}[Y; \boldsymbol{X}_{\boldsymbol{\alpha}}] p_{\mathcal{A}_{\boldsymbol{\gamma}}}(\boldsymbol{\alpha}) \tag{33}$$

$$= \sum_{\boldsymbol{\alpha} \in \{0,1\}^T} \mathbb{I}[Y; \boldsymbol{X}_{\boldsymbol{\alpha}}] \prod_{t=1}^{T} (1-\gamma_t)^{\alpha_t} \gamma_t^{1-\alpha_t} \tag{34}$$

is continuous in $\boldsymbol{\gamma}$. Clearly, $\mathcal{L}_{\text{ideal}}$ is also continuous in $\boldsymbol{\gamma}$, and by the extreme value theorem there must be some vector $\boldsymbol{\gamma}^* \in [0,1]^T$ such that $\mathcal{L}_{\text{ideal}}(\boldsymbol{\gamma}^*) = \inf_{\boldsymbol{\gamma} \in [0,1]^T} \mathcal{L}(\boldsymbol{\gamma})$.

We now derive an implicit expression that such $\boldsymbol{\gamma}^*$ must satisfy. Note that by equation 31, we can write

$$\mathcal{L}_{\text{ideal}}(\boldsymbol{\gamma}) = \left[ \frac{1}{2}\lambda \left\| \boldsymbol{\gamma}_{-t} \right\|_2^2 + \sum_{\substack{\boldsymbol{\alpha} \in \{0,1\}^T \\ \alpha_t = 0}} \mathbb{I}[Y; X_t, \boldsymbol{X_\alpha}] p_{\boldsymbol{\mathcal{A}}_{\gamma,-t}}(\boldsymbol{\alpha}_{-t}) \right]$$

$$- \gamma_t \left[ \sum_{\substack{\boldsymbol{\alpha} \in \{0,1\}^T \\ \alpha_t = 0}} \mathbb{I}[Y; X_t \mid \boldsymbol{X_\alpha}] p_{\boldsymbol{\mathcal{A}}_{\gamma,-t}}(\boldsymbol{\alpha}_{-t}) \right] + \frac{1}{2}\lambda \gamma_t^2 \tag{35}$$

$$=: f_1(\boldsymbol{\gamma}_{-t}) - \gamma_t f_2(\boldsymbol{\gamma}_{-t}) + \frac{1}{2}\lambda \gamma_t^2. \tag{36}$$

Our optimization problem may thus be expressed

$$\min_{\boldsymbol{\gamma} \in [0,1]^T} \mathcal{L}_{\text{ideal}}(\boldsymbol{\gamma}) \tag{37}$$

$$\equiv \min_{\boldsymbol{\gamma}_{-t} \in [0,1]^{T-1}} \min_{\gamma_t \in [0,1]} \frac{1}{2}\lambda \gamma_t^2 - \gamma_t f_2(\boldsymbol{\gamma}_{-t}) + f_1(\boldsymbol{\gamma}_{-t}). \tag{38}$$

Consider the inner optimization problem and observe that

$$\frac{\partial}{\partial \gamma_t} \mathcal{L}_{\text{ideal}}(\gamma_t, \boldsymbol{\gamma}_{-t}) = \lambda \gamma_t - f_2(\boldsymbol{\gamma}_{-t}) \quad \text{and} \quad \frac{\partial^2}{\partial \gamma_t^2} \mathcal{L}_{\text{ideal}}(\gamma_t, \boldsymbol{\gamma}_{-t}) = \lambda > 0. \tag{39}$$

It follows that if $\frac{f_2(\boldsymbol{\gamma}_{-t})}{\lambda} \in [0,1]$, then it is the sole element of $\min_{\gamma_t \in [0,1]} \mathcal{L}_{\text{ideal}}(\gamma_t, \boldsymbol{\gamma}_{-t})$. Because $\lambda > 0$ and $f_2(\cdot) \geq 0$, we can never have $\frac{f_2(\boldsymbol{\gamma}_{-t})}{\lambda} < 0$. If $\frac{f_2(\boldsymbol{\gamma}_{-t})}{\lambda} > 1$, then $\frac{\partial}{\partial \gamma_t} \mathcal{L}_{\text{ideal}}(\gamma_t, \boldsymbol{\gamma}_{-t}) < 0$ for all $\gamma_t \in [0,1]$, which implies that $1$ is the sole element of $\arg\min_{\gamma_t \in [0,1]} \mathcal{L}_{\text{ideal}}(\gamma_t, \boldsymbol{\gamma}_{-t})$. The implicit form for $\gamma_t^*$ listed above follows from replacing $p_{\boldsymbol{\mathcal{A}}_{\gamma,-t}}(\boldsymbol{\alpha}_{-t})$ by its definition. $\qquad \square$

**Corrolary 1.1.** *Under the conditions of Proposition 1, if $\lambda > \log|\mathsf{Y}|$, then $\boldsymbol{\gamma}^* \in [0,1)^T$.*

*Proof.* Note that for arbitrary $t \in [1 \, .. \, T]$ and $\mathsf{S} \subset \{X_1, \ldots, X_T\} \setminus \{X_t\}$, we have the inequality

$$\mathbb{I}[Y; X_t \mid \mathsf{S}] = \mathbb{H}[Y \mid \mathsf{S}] - \mathbb{H}[Y \mid X_t, \mathsf{S}] \tag{40}$$

$$\leq \mathbb{H}[Y \mid \mathsf{S}] \tag{41}$$

$$\leq \mathbb{H}[Y] \tag{42}$$

$$\leq \log|\mathsf{Y}|. \tag{43}$$

Thus, $\gamma_t^* < 1$ provided

$$\frac{1}{\lambda} \sum_{\substack{\boldsymbol{\alpha} \in \{0,1\}^T \\ \alpha_t = 0}} \mathbb{I}[Y; X_t \mid \boldsymbol{X_\alpha}] p_{\boldsymbol{\mathcal{A}}_{\gamma,-t}}(\boldsymbol{\alpha}_{-t}) \tag{44}$$

$$\leq \frac{\log|\mathsf{Y}|}{\lambda} \sum_{\substack{\boldsymbol{\alpha} \in \{0,1\}^T \\ \alpha_t = 0}} p_{\boldsymbol{\mathcal{A}}_{\gamma,-t}}(\boldsymbol{\alpha}_{-t}) \tag{45}$$

$$= \frac{\log|\mathsf{Y}|}{\lambda} < 1. \tag{46}$$

$$\square$$

**Corrolary 1.2.** *Suppose the conditions of Proposition 1 are satisfied. Consider $X_t$ such that $\mathbb{I}[Y; X_t \mid S] = 0$ for all sets $\mathsf{S} \subset \{X_1, \ldots, X_T\} \setminus \{X_t\}$. It follows immediately from Proposition 1 that $\gamma_t^* = 0$. Suppose it additionally holds that $\mathbb{I}[Y; X_t \mid \mathsf{S}] > 0 \implies \mathbb{I}[Y; X_t \mid X_\tau, \mathsf{S}] > 0$ for all $t, \tau \in [1 \, .. \, T]$ such that $t \neq \tau$ and for all $\mathsf{S} \subset \{X_1, \ldots, X_T\} \setminus \{X_t, X_\tau\}$. Let $\lambda$ be sufficiently-large that $\boldsymbol{\gamma}^* \in [0,1)^T$. If there exists some $\mathsf{S} \subset \{X_1, \ldots, X_T\} \setminus \{X_t\}$ such that $\mathbb{I}[Y; X_t \mid \mathsf{S}] > 0$, then $\gamma_t^* > 0$.*

*Proof.* Recall that by Corollary 1.1, there exists some finite $\lambda$ for which $\boldsymbol{\gamma}^* \in [0,1)^T$. It follows that we can write

$$\gamma_t^* = \frac{1}{\lambda} \sum_{\substack{\boldsymbol{\alpha} \in \{0,1\}^T \\ \alpha_t = 0}} \mathbb{I}[Y; X_t \mid \boldsymbol{X_\alpha}] \prod_{\tau \in [1\,..\,T] \setminus \{t\}} (\gamma_\tau^*)^{1-\alpha_\tau} (1 - \gamma_\tau^*)^{\alpha_\tau} \tag{47}$$

$$\geq \frac{1}{\lambda} \mathbb{I}[Y; X_t \mid \boldsymbol{X_{\alpha'}}] \prod_{\tau \in [1\,..\,T] \setminus \{t\}} (\gamma_\tau^*)^{1-\alpha'_\tau} (1 - \gamma_\tau^*)^{\alpha'_\tau} \tag{48}$$

where $\boldsymbol{\alpha'} := (1 \text{ if } X_t \in \mathsf{S} \text{ else } 0 : t = 1, \dots, T)$. If this quantity is greater than zero, then our claim is satisfied. Else, suppose it is equal to zero. Since we have assumed that $\mathbb{I}[Y; X_t \mid \boldsymbol{X_{\alpha'}}] > 0$ and $\gamma_\tau < 1 \; \forall \tau$, it must be the case that $\gamma_\tau^* = \alpha'_\tau = 0$ for at least one $\tau$.

Suppose we have timesteps $\{\tau_i : i \in [1\,..\,\eta]\}$ for which $\gamma_{\tau_i}^* = \alpha'_{\tau_i} = 0$. Consider the vector $\tilde{\boldsymbol{\alpha}} := (1 \text{ if } \alpha'_\tau = 1 \text{ or } \tau = \tau_i \text{ for some } i \in [1\,..\,\eta])$, i.e. $\boldsymbol{\alpha'}$ with all of its 'offending' 0's flipped to 1's. Note that $\prod_{\tau \in [1\,..\,T] \setminus \{t\}} (\gamma_\tau^*)^{1-\tilde{\alpha}_\tau} (1 - \gamma_\tau^*)^{\tilde{\alpha}_\tau} > 0$. Furthermore, by assumption we have

$$\mathbb{I}[Y; X_t \mid \boldsymbol{X_{\alpha'}}] > 0 \tag{49}$$

$$\implies \quad \mathbb{I}[Y; X_t \mid \boldsymbol{X_{\alpha'}}, X_{\tau_1}] > 0 \tag{50}$$

$$\implies \cdots \implies \quad \mathbb{I}[Y; X_t \mid \boldsymbol{X_{\alpha'}}, X_{\tau_1}, \dots, X_{\tau_\eta}] = \mathbb{I}[Y; X_t \mid \boldsymbol{X_{\tilde{\alpha}}}] > 0. \tag{51}$$

Thus, we have

$$\gamma_t^* \geq \frac{1}{\lambda} \mathbb{I}[Y; X_t \mid \boldsymbol{X_{\tilde{\alpha}}}] \prod_{\tau \in [1\,..\,T] \setminus \{t\}} (\gamma_\tau^*)^{1-\tilde{\alpha}_\tau} (1 - \gamma_\tau^*)^{\tilde{\alpha}_\tau} > 0. \tag{52}$$

$\square$

These results suggest that we should choose $\lambda$ to be large enough that no $\gamma_t^*$ saturates at 1, and that for consistency we should quantify leakage using $\lambda \boldsymbol{\gamma}^*$ rather than $\boldsymbol{\gamma}^*$.

## B.4 EQUIVALENT ADVERSARIAL GAME

In practice we cannot solve equation 27 directly because we lack an expression for $p_{\boldsymbol{X}, Y}$. Here we will propose a different optimization problem which is equivalent to equation 27, and allows us to use deep learning to characterize $p_{\boldsymbol{X}, Y}$ using data.

Consider the family $\Phi := \{\Phi_{\boldsymbol{\alpha}}\}_{\boldsymbol{\alpha} \in \{0,1\}^T}$ with each element a deep neural network

$$\Phi_{\boldsymbol{\alpha}} : \mathsf{Y} \times \mathbb{R}^{\sum_{t=1}^T \alpha_t} \times \mathbb{R}^P \to \mathbb{R}_+ : (y, \boldsymbol{x}, \boldsymbol{\theta}) \mapsto \Phi_{\boldsymbol{\alpha}}(y \mid \boldsymbol{x}; \boldsymbol{\theta}). \tag{53}$$

We denote by $\Phi_{\boldsymbol{\alpha}}(y \mid \boldsymbol{x}; \boldsymbol{\theta})$ the mass assigned to $y$ by the network $\Phi_{\boldsymbol{\alpha}}$ with weights $\boldsymbol{\theta}$ and input $\boldsymbol{x}$. We assume each $\Phi_{\boldsymbol{\alpha}}(\cdot \mid \boldsymbol{x}; \boldsymbol{\theta})$ is a probability mass function over $\mathsf{Y}$ (e.g. the neural net has a softmax output activation). Consider the optimization problem

$$\min_{\boldsymbol{\gamma} \in [0,1]^T} \max_{\boldsymbol{\theta} \in \mathbb{R}^P} \quad \mathcal{L}_{\text{adv}}(\boldsymbol{\gamma}, \boldsymbol{\theta}) := \frac{1}{2} \lambda \|\boldsymbol{\gamma}\|_2^2 + \mathbb{E} \log \Phi_{\boldsymbol{\mathcal{A}_\gamma}}(Y \mid \boldsymbol{X_{\mathcal{A}_\gamma}}; \boldsymbol{\theta}). \tag{54}$$

**Proposition 2.** *Consider the objective function $\mathcal{L}_{adv}$ of equation 54. Suppose there exists some $\boldsymbol{\theta}^* \in \mathbb{R}^P$ such that $\Phi_{\boldsymbol{\alpha}}(y \mid \boldsymbol{x_\alpha}; \boldsymbol{\theta}^*) = p_{Y \mid \boldsymbol{X_\alpha}}(y \mid \boldsymbol{x_\alpha})$ for all $\boldsymbol{\alpha} \in \{0,1\}^T, \boldsymbol{x} \in \mathbb{R}^T, y \in \mathsf{Y}$. Then*

$$\boldsymbol{\theta}^* \in \arg \max_{\boldsymbol{\theta} \in \mathbb{R}^P} \mathcal{L}_{adv}(\boldsymbol{\gamma}, \boldsymbol{\theta}) \quad \forall \boldsymbol{\gamma} \in [0,1]^T. \tag{55}$$

*Furthermore, for all $y \in \mathsf{Y}$ and for all $\boldsymbol{\gamma} \in [0,1]^T, \alpha \in \{0,1\}^T$ such that $p_{\boldsymbol{\mathcal{A}_\gamma}}(\boldsymbol{\alpha}) > 0$,*

$$\Phi_{\boldsymbol{\alpha}}(y \mid \boldsymbol{X_\alpha}; \hat{\boldsymbol{\theta}}) = p_{Y \mid \boldsymbol{X_\alpha}}(y \mid \boldsymbol{X_\alpha}) \quad p_{\boldsymbol{X}}\text{-almost surely} \quad \forall \hat{\boldsymbol{\theta}} \in \arg \min_{\boldsymbol{\theta} \in \mathbb{R}^P} \mathcal{L}_{adv}(\boldsymbol{\gamma}, \boldsymbol{\theta}). \tag{56}$$

*Proof.* Note that since each $\Phi_{\boldsymbol{\alpha}}(\cdot \mid \boldsymbol{x}, \boldsymbol{\theta})$ is a probability mass function over $\mathsf{Y}$, by Gibbs' inequality we have that

$$\mathbb{E} \log \Phi_{\boldsymbol{\alpha}}(Y \mid \boldsymbol{X_\alpha}; \boldsymbol{\theta}) \leq \mathbb{E} \log p_{Y \mid \boldsymbol{X_\alpha}}(Y \mid \boldsymbol{X_\alpha}) \quad \forall \boldsymbol{\alpha} \in \{0,1\}^T, \boldsymbol{\theta} \in \mathbb{R}^P. \tag{57}$$

Thus,

$$\mathcal{L}_{\text{adv}}(\boldsymbol{\gamma}, \boldsymbol{\theta}^*) \geq \mathcal{L}_{\text{adv}}(\boldsymbol{\gamma}, \boldsymbol{\theta}) \quad \forall \boldsymbol{\theta} \in \mathbb{R}^P, \ \boldsymbol{\gamma} \in [0,1]^T, \tag{58}$$

which implies the first claim.

Next, consider some fixed $\boldsymbol{\gamma} \in [0,1]^T$ and $\hat{\boldsymbol{\theta}} \in \arg\min_{\boldsymbol{\theta} \in \mathbb{R}^P} \mathcal{L}_{\text{adv}}(\boldsymbol{\gamma}, \boldsymbol{\theta})$. We must have $\mathcal{L}_{\text{adv}}(\boldsymbol{\gamma}, \hat{\boldsymbol{\theta}}) = \mathcal{L}_{\text{adv}}(\boldsymbol{\gamma}, \boldsymbol{\theta}^*)$. Thus,

$$0 = \mathcal{L}_{\text{adv}}(\boldsymbol{\gamma}, \hat{\boldsymbol{\theta}}) - \mathcal{L}_{\text{adv}}(\boldsymbol{\gamma}, \boldsymbol{\theta}^*) \tag{59}$$

$$= \mathbb{E}\left[\log p_{Y|\boldsymbol{X}_{\boldsymbol{\alpha}}}(Y \mid \boldsymbol{X}_{\boldsymbol{\alpha}}) - \log \Phi_{\boldsymbol{\alpha}}(Y \mid \boldsymbol{X}_{\boldsymbol{\alpha}}; \hat{\boldsymbol{\theta}})\right] \tag{60}$$

$$= \sum_{\boldsymbol{\alpha} \in \{0,1\}^T} p_{\mathcal{A}_{\boldsymbol{\gamma}}}(\boldsymbol{\alpha}) \, \mathbb{E}\left[\log p_{Y|\boldsymbol{X}_{\boldsymbol{\alpha}}}(Y \mid \boldsymbol{X}_{\boldsymbol{\alpha}}) - \log \Phi_{\boldsymbol{\alpha}}(Y \mid \boldsymbol{X}_{\boldsymbol{\alpha}}; \hat{\boldsymbol{\theta}})\right]. \tag{61}$$

By Gibbs' inequality each of the expectations in the summation is nonnegative, which implies that whenever $p_{\mathcal{A}_{\boldsymbol{\gamma}}}(\boldsymbol{\alpha}) > 0$ we must have

$$0 = \mathbb{E}\left[\log p_{Y|\boldsymbol{X}_{\boldsymbol{\alpha}}}(Y \mid \boldsymbol{X}_{\boldsymbol{\alpha}}) - \log \Phi_{\boldsymbol{\alpha}}(Y \mid \boldsymbol{X}_{\boldsymbol{\alpha}}; \hat{\boldsymbol{\theta}})\right] \tag{62}$$

$$= \int_{\mathbb{R}^{\sum_{t=1}^T \alpha_t}} p_{\boldsymbol{X}_{\boldsymbol{\alpha}}}(\boldsymbol{x}_{\boldsymbol{\alpha}}) \, \mathbb{KL}\left[p_{Y|\boldsymbol{X}_{\boldsymbol{\alpha}}}(\cdot \mid \boldsymbol{x}_{\boldsymbol{\alpha}}) \,\|\, \Phi_{\boldsymbol{\alpha}}(\cdot \mid \boldsymbol{x}_{\boldsymbol{\alpha}}; \hat{\boldsymbol{\theta}})\right] d\boldsymbol{x}_{\boldsymbol{\alpha}}. \tag{63}$$

Since $\mathbb{KL}\left[p_{Y|\boldsymbol{X}_{\boldsymbol{\alpha}}}(\cdot \mid \boldsymbol{x}_{\boldsymbol{\alpha}}) \,\|\, \Phi_{\boldsymbol{\alpha}}(\cdot \mid \boldsymbol{x}_{\boldsymbol{\alpha}}; \hat{\boldsymbol{\theta}})\right] \geq 0$ with equality if and only if $p_{Y|\boldsymbol{X}_{\boldsymbol{\alpha}}}(y \mid \boldsymbol{x}_{\boldsymbol{\alpha}}) = \Phi_{\boldsymbol{\alpha}}(y \mid \boldsymbol{x}_{\boldsymbol{\alpha}}; \hat{\boldsymbol{\theta}}) \ \forall y \in \mathsf{Y}$, this must be the case except possibly for $\boldsymbol{x} \in \mathbb{R}^T$ where

$$\int_{\{\boldsymbol{x}_{\boldsymbol{\alpha}} : \boldsymbol{x} \in \mathbb{R}^T\}} p_{\boldsymbol{X}_{\boldsymbol{\alpha}}}(\boldsymbol{x}_{\boldsymbol{\alpha}}) \, d\boldsymbol{x}_{\boldsymbol{\alpha}} = 0 \implies \int_{\mathbb{R}^T} p_{\boldsymbol{X}}(\boldsymbol{x}) \, d\boldsymbol{x} = 0. \tag{64}$$

This implies the second claim. $\qquad \square$

**Corollary 2.1.** *Under the assumptions of Proposition 2, equations 54 and 27 are equivalent.*

*Proof.* Observe

$$\min_{\boldsymbol{\gamma} \in [0,1]^T} \max_{\boldsymbol{\theta} \in \mathbb{R}^P} \quad \mathcal{L}_{\text{adv}}(\boldsymbol{\gamma}, \boldsymbol{\theta}) \tag{65}$$

$$\equiv \min_{\boldsymbol{\gamma} \in [0,1]^T} \max_{\boldsymbol{\theta} \in \mathbb{R}^P} \quad \frac{1}{2}\lambda \|\boldsymbol{\gamma}\|_2^2 + \mathbb{E} \log \Phi_{\mathcal{A}_{\boldsymbol{\gamma}}}(Y \mid \boldsymbol{X}_{\mathcal{A}_{\boldsymbol{\gamma}}}; \boldsymbol{\theta}) \tag{66}$$

$$\equiv \min_{\boldsymbol{\gamma} \in [0,1]^T} \quad \frac{1}{2}\lambda \|\boldsymbol{\gamma}\|_2^2 + \sum_{\boldsymbol{\alpha} \in \{0,1\}^T} p_{\mathcal{A}_{\boldsymbol{\gamma}}}(\boldsymbol{\alpha}) \, \mathbb{E} \log p_{Y|\boldsymbol{X}_{\boldsymbol{\alpha}}}(Y \mid \boldsymbol{X}_{\boldsymbol{\alpha}}) \tag{67}$$

$$\quad \text{by Proposition 2}$$

$$\equiv \min_{\boldsymbol{\gamma} \in [0,1]^T} \quad \frac{1}{2}\lambda \|\boldsymbol{\gamma}\|_2^2 - \sum_{\boldsymbol{\alpha} \in \{0,1\}^T} p_{\mathcal{A}_{\boldsymbol{\gamma}}}(\boldsymbol{\alpha}) \, \mathbb{H}[Y \mid \boldsymbol{X}_{\boldsymbol{\alpha}}] \tag{68}$$

$$\equiv \min_{\boldsymbol{\gamma} \in [0,1]^T} \quad \frac{1}{2}\lambda \|\boldsymbol{\gamma}\|_2^2 + \sum_{\boldsymbol{\alpha} \in \{0,1\}^T} p_{\mathcal{A}_{\boldsymbol{\gamma}}}(\boldsymbol{\alpha}) \, [\mathbb{H}[Y] - \mathbb{H}[Y \mid \boldsymbol{X}_{\boldsymbol{\alpha}}]] \tag{69}$$

$$\equiv \min_{\boldsymbol{\gamma} \in [0,1]^T} \quad \frac{1}{2}\lambda \|\boldsymbol{\gamma}\|_2^2 + \sum_{\boldsymbol{\alpha} \in \{0,1\}^T} p_{\mathcal{A}_{\boldsymbol{\gamma}}}(\boldsymbol{\alpha}) \, \mathbb{I}[Y; \boldsymbol{X}_{\boldsymbol{\alpha}}] \tag{70}$$

$$\equiv \min_{\boldsymbol{\gamma} \in [0,1]^T} \quad \frac{1}{2}\lambda \|\boldsymbol{\gamma}\|_2^2 + \mathbb{I}[Y; \boldsymbol{X}_{\mathcal{A}_{\boldsymbol{\gamma}}} \mid \mathcal{A}_{\boldsymbol{\gamma}}] \tag{71}$$

$$\equiv \min_{\boldsymbol{\gamma} \in [0,1]^T} \quad \mathcal{L}_{\text{ideal}}(\boldsymbol{\gamma}). \tag{72}$$

$$\square$$

**Corollary 2.2.** *Suppose the conditions of Proposition 2 are satisfied, and let $\hat{\boldsymbol{\theta}} \in \arg\min_{\boldsymbol{\theta} \in \mathbb{R}^P} \mathcal{L}_{\text{adv}}(\boldsymbol{\gamma}, \boldsymbol{\theta})$ for some $\boldsymbol{\gamma} \in [0,1]^T$. Consider $\boldsymbol{\alpha} := \boldsymbol{\alpha}' + \boldsymbol{\alpha}''$ where $\boldsymbol{\alpha}', \boldsymbol{\alpha}'' \in \{0,1\}^T$ such that $\alpha_t' = 1 \implies \alpha_t'' = 0$, $\alpha_t'' = 1 \implies \alpha_t' = 0$, and $p_{\mathcal{A}_{\boldsymbol{\gamma}}}(\boldsymbol{\alpha}) > 0$. For all $y \in \mathsf{Y}$, almost*

*surely for $\boldsymbol{X} \sim p_{\boldsymbol{X}}$, it follows immediately from Proposition 2 that we can compute the conditional pointwise mutual information as*

$$\mathrm{pmi}(y; \boldsymbol{X}_{\boldsymbol{\alpha}'} \mid \boldsymbol{X}_{\boldsymbol{\alpha}''}) := \log p_{Y|\boldsymbol{X}_{\boldsymbol{\alpha}}}(y \mid \boldsymbol{X}_{\boldsymbol{\alpha}}) - \log p_{Y|\boldsymbol{X}_{\boldsymbol{\alpha}''}}(y \mid \boldsymbol{X}_{\boldsymbol{\alpha}''}) \tag{73}$$

$$= \log \Phi(y \mid \boldsymbol{\alpha} \odot \boldsymbol{X}, \boldsymbol{\alpha}; \hat{\boldsymbol{\theta}}) - \log \Phi(y \mid \boldsymbol{\alpha}'' \odot \boldsymbol{X}, \boldsymbol{\alpha}''; \hat{\boldsymbol{\theta}}). \tag{74}$$

This property is useful because it allows us to assess the leakage of a *single* power trace, as opposed to merely a summary of the entire distribution of traces. There are scenarios where a point might leak for some traces but not for others. For example, a common countermeasure is to randomly delay leaky instructions or swap their order with another instruction so that they do not occur at a deterministic time relative to the start of encryption.

## B.5 IMPLEMENTATION DETAILS

It would be impractical to train $2^T$ neural nets independently, so we amortize the cost by instead training a single network with $\boldsymbol{\alpha}$ as an auxiliary input:

$$\Phi : \mathsf{Y} \times \mathbb{R}^T \times \{0,1\}^T \times \mathbb{R}^P : (y, \tilde{\boldsymbol{x}}, \boldsymbol{\alpha}, \boldsymbol{\theta}) \mapsto \Phi(y \mid \tilde{\boldsymbol{x}}, \boldsymbol{\alpha}, \boldsymbol{\theta}) \tag{75}$$

where each $\Phi_{\boldsymbol{\alpha}}(y \mid \boldsymbol{x}_{\boldsymbol{\alpha}}; \boldsymbol{\theta}) := \Phi(y \mid \boldsymbol{\alpha} \odot \boldsymbol{x}, \boldsymbol{\alpha}; \boldsymbol{\theta})$. We can then re-write equation 54 as

$$\min_{\boldsymbol{\gamma} \in [0,1]^T} \max_{\boldsymbol{\theta} \in \mathbb{R}^P} \quad \mathcal{L}(\boldsymbol{\gamma}, \boldsymbol{\theta}) := \frac{1}{2}\lambda \|\boldsymbol{\gamma}\|_2^2 + \mathbb{E} \log \Phi(Y \mid \mathcal{A}_{\boldsymbol{\gamma}} \odot \boldsymbol{X}, \mathcal{A}_{\boldsymbol{\gamma}}; \boldsymbol{\theta}). \tag{76}$$

We intend to approximately solve equation 76 using an alternating minibatch stochastic gradient descent-style technique, similarly to GANs (Goodfellow et al., 2014). To do so, we must first convert it into an equivalent unconstrained optimization problem so that it is amenable to gradient descent. We must then derive expressions for the gradients of our objective function which can be estimated using automatic differentiation (Paszke et al., 2019) and Monte Carlo integration.

Note that the inner optimization problem is already unconstrained, and it is immediate that

$$\nabla_{\boldsymbol{\theta}} \mathcal{L}(\boldsymbol{\gamma}, \boldsymbol{\theta}) = \mathbb{E} \, \nabla_{\boldsymbol{\theta}} \log \Phi(Y \mid \mathcal{A}_{\boldsymbol{\gamma}} \odot \boldsymbol{X}, \mathcal{A}_{\boldsymbol{\gamma}}; \boldsymbol{\theta}). \tag{77}$$

We can re-parameterize the outer problem and derive an appropriate gradient expression as follows:

**Proposition 3.** *The optimization problem of equation 76 is equivalent to*

$$\min_{\boldsymbol{\gamma}' \in (\mathbb{R} \cup \{\pm\infty\})^T} \max_{\boldsymbol{\theta} \in \mathbb{R}^P} \quad \mathcal{L}(\mathrm{Sigmoid}(\boldsymbol{\gamma}', \boldsymbol{\theta}) \tag{78}$$

*where $\boldsymbol{\gamma} = \mathrm{Sigmoid}(\boldsymbol{\gamma}')$. Furthermore, we can express*

$$\nabla_{\boldsymbol{\gamma}'} \mathcal{L}(\boldsymbol{\gamma}, \boldsymbol{\theta}) = \lambda \boldsymbol{\gamma} \odot \boldsymbol{\gamma} \odot \overline{\boldsymbol{\gamma}} + \mathbb{E} \left[ \log \Phi(Y \mid \mathcal{A}_{\boldsymbol{\gamma}} \odot \boldsymbol{X}, \mathcal{A}_{\boldsymbol{\gamma}}; \boldsymbol{\theta}) \cdot (\overline{\mathcal{A}}_{\boldsymbol{\gamma}} \odot \overline{\boldsymbol{\gamma}} - \mathcal{A}_{\boldsymbol{\gamma}} \odot \boldsymbol{\gamma}) \right], \tag{79}$$

*where for compactness we have left implicit that $\boldsymbol{\gamma}$ is a function of $\boldsymbol{\gamma}'$, and have denoted $\overline{\mathcal{A}}_{\boldsymbol{\gamma}} := \mathbf{1} - \mathcal{A}_{\boldsymbol{\gamma}}$ and $\overline{\boldsymbol{\gamma}} := \mathbf{1} - \boldsymbol{\gamma}$.*

*Proof.* Note that because Sigmoid is bijective and $\mathrm{Sigmoid}\left((\mathbb{R} \cup \{\pm\infty\})^T\right) = [0,1]^T$, by the change of variables formula of Boyd & Vandenberghe (2004, pp. 130),

$$\min_{\boldsymbol{\gamma} \in [0,1]^T} \max_{\boldsymbol{\theta} \in \mathbb{R}^P} \quad \mathcal{L}(\boldsymbol{\gamma}, \boldsymbol{\theta}) \quad \equiv \quad \min_{\boldsymbol{\gamma}' \in (\mathbb{R} \cup \{\pm\infty\})^T} \max_{\boldsymbol{\theta} \in \mathbb{R}^P} \quad \mathcal{L}(\mathrm{Sigmoid}(\boldsymbol{\gamma}'), \boldsymbol{\theta}). \tag{80}$$

Let us now derive an expression for $\nabla_{\boldsymbol{\gamma}'} \mathcal{L}(\boldsymbol{\gamma} = \mathrm{Sigmoid}(\boldsymbol{\gamma}'), \boldsymbol{\theta})$, considering the two terms of $\mathcal{L}$ separately. Consider the leftmost term and note that for all $t \in [1 .. T]$,

$$\frac{\partial}{\partial \gamma_t'} \frac{1}{2} \|\boldsymbol{\gamma}\|_2^2 \quad = \quad \gamma_t \frac{d\gamma_t}{d\gamma_t'} \tag{81}$$

$$= \quad \gamma_t^2 (1 - \gamma_t) \tag{82}$$

$$\implies \nabla_{\boldsymbol{\gamma}'} \frac{1}{2} \|\boldsymbol{\gamma}\|_2^2 \quad = \quad \boldsymbol{\gamma} \odot \boldsymbol{\gamma} \odot \overline{\boldsymbol{\gamma}}. \tag{83}$$

Now considering the rightmost term, observe that because $\boldsymbol{\mathcal{A}_\gamma}$ is independent of $\boldsymbol{X}$ and $Y$,

$$\nabla_{\gamma'} \mathbb{E} \log \Phi \left( Y \mid \boldsymbol{\mathcal{A}_\gamma} \odot \boldsymbol{X}, \boldsymbol{\mathcal{A}_\gamma}; \boldsymbol{\theta} \right) \tag{84}$$

$$= \nabla_{\gamma'} \mathbb{E}_{\boldsymbol{X},Y} \mathbb{E}_{\boldsymbol{\mathcal{A}_\gamma}} \log \Phi \left( Y \mid \boldsymbol{\mathcal{A}_\gamma} \odot \boldsymbol{X}, \boldsymbol{\mathcal{A}_\gamma}; \boldsymbol{\theta} \right) \tag{85}$$

$$= \mathbb{E}_{\boldsymbol{X},Y} \nabla_{\gamma'} \mathbb{E}_{\boldsymbol{\mathcal{A}_\gamma}} \log \Phi \left( Y \mid \boldsymbol{\mathcal{A}_\gamma} \odot \boldsymbol{X}, \boldsymbol{\mathcal{A}_\gamma}; \boldsymbol{\theta} \right). \tag{86}$$

Using the REINFORCE estimator (Williams, 1992), we can bring the gradient operator into the expected value. For arbitrary $\boldsymbol{x} \in \mathbb{R}^T$, $\boldsymbol{y} \in \mathsf{Y}$, $\boldsymbol{\theta} \in \mathbb{R}^P$,

$$\nabla_{\gamma'} \mathbb{E}_{\boldsymbol{\mathcal{A}_\gamma}} \log \Phi(y \mid \boldsymbol{\mathcal{A}_\gamma} \odot \boldsymbol{x}, \boldsymbol{\mathcal{A}_\gamma}; \boldsymbol{\theta}) \tag{87}$$

$$= \nabla_{\gamma'} \sum_{\boldsymbol{\alpha} \in \{0,1\}^T} \log \Phi(y \mid \boldsymbol{\alpha} \odot \boldsymbol{x}, \boldsymbol{\alpha}; \boldsymbol{\theta}) \cdot p_{\boldsymbol{\mathcal{A}_\gamma}}(\boldsymbol{\alpha}) \tag{88}$$

$$= \sum_{\boldsymbol{\alpha} \in \{0,1\}^T} \log \Phi(y \mid \boldsymbol{\alpha} \odot \boldsymbol{x}, \boldsymbol{\alpha}; \boldsymbol{\theta}) \cdot \nabla_{\gamma'} p_{\boldsymbol{\mathcal{A}_\gamma}}(\boldsymbol{\alpha}) \tag{89}$$

$$= \sum_{\boldsymbol{\alpha} \in \{0,1\}^T} \log \Phi(y \mid \boldsymbol{\alpha} \odot \boldsymbol{x}, \boldsymbol{\alpha}; \boldsymbol{\theta}) \cdot p_{\boldsymbol{\mathcal{A}_\gamma}}(\boldsymbol{\alpha}) \nabla_{\gamma'} \log p_{\boldsymbol{\mathcal{A}_\gamma}}(\boldsymbol{\alpha}) \tag{90}$$

$$= \mathbb{E}_{\boldsymbol{\mathcal{A}_\gamma}} \left[ \log \Phi(y \mid \boldsymbol{\mathcal{A}_\gamma} \odot \boldsymbol{x}, \boldsymbol{\mathcal{A}_\gamma}; \boldsymbol{\theta}) \cdot \nabla_{\gamma'} \log p_{\boldsymbol{\mathcal{A}_\gamma}}(\boldsymbol{\mathcal{A}_\gamma}) \right]. \tag{91}$$

Note that we can express

$$p_{\boldsymbol{\mathcal{A}_\gamma}}(\boldsymbol{\alpha}) = \prod_{t=1}^{T} \gamma_t^{1-\alpha_t}(1-\gamma_t)^{\alpha_t} \tag{92}$$

and observe that for arbitrary $t \in [1 .. T]$, $\boldsymbol{\alpha} \in \{0,1\}^T$,

$$\frac{\partial}{\partial \gamma_t'} \log p_{\boldsymbol{\mathcal{A}_\gamma}}(\boldsymbol{\alpha}) = \frac{\partial}{\partial \gamma_t'} \log \left( \prod_{\tau=1}^{T} (1-\gamma_\tau)^{\alpha_\tau} \gamma_\tau^{1-\alpha_\tau} \right) \tag{93}$$

$$= \frac{\partial}{\partial \gamma_t'} \sum_{\tau=1}^{T} \alpha_\tau \log(1-\gamma_\tau) + (1-\alpha_\tau) \log \gamma_\tau \tag{94}$$

$$= -\frac{d\gamma_t}{d\gamma_t'} \frac{\alpha_t}{1-\gamma_t} + \frac{d\gamma_t}{d\gamma_t'} \frac{1-\alpha_t}{\gamma_t} \tag{95}$$

$$= (1-\alpha_t)(1-\gamma_t) - \alpha_t \gamma_t \tag{96}$$

$$\implies \nabla_{\gamma'} p_{\boldsymbol{\mathcal{A}_\gamma}}(\boldsymbol{\alpha}) = \overline{\boldsymbol{\alpha}} \odot \overline{\boldsymbol{\gamma}} - \boldsymbol{\alpha} \odot \boldsymbol{\gamma}. \tag{97}$$

Combining these results, we get

$$\nabla_{\gamma'} \mathcal{L}(\boldsymbol{\gamma}, \boldsymbol{\theta}) = \lambda \boldsymbol{\gamma} \odot \boldsymbol{\gamma} \odot \overline{\boldsymbol{\gamma}} + \mathbb{E} \left[ \log \Phi \left( Y \mid \boldsymbol{\mathcal{A}_\gamma} \odot \boldsymbol{X}, \boldsymbol{\mathcal{A}_\gamma}; \boldsymbol{\theta} \right) \cdot \left( \overline{\boldsymbol{\mathcal{A}}}_\gamma \odot \overline{\boldsymbol{\gamma}} - \boldsymbol{\mathcal{A}_\gamma} \odot \boldsymbol{\gamma} \right) \right]. \tag{98}$$

$$\square$$

In practice, when computing Monte Carlo estimates of this gradient we find it helpful to subtract $\hat{\ell}$ from $\log \Phi \left( Y \mid \boldsymbol{\mathcal{A}_\gamma} \odot \boldsymbol{X}, \boldsymbol{\mathcal{A}_\gamma}; \boldsymbol{\theta} \right)$, where $\hat{\ell}$ is an exponentially-weighted moving average of $\log \Phi \left( Y \mid \boldsymbol{\mathcal{A}_\gamma} \odot \boldsymbol{X}, \boldsymbol{\mathcal{A}_\gamma}; \boldsymbol{\theta} \right)$ updated after every training step. This is a standard technique from reinforcement learning which reduces the variance of the gradient estimator for practically no cost. Note that because $\hat{\ell}$ does not depend on $\boldsymbol{\gamma}$, it does not change the expected value of our gradient estimator:

$$\nabla_{\gamma} \mathbb{E} \left[ \log \Phi \left( Y \mid \boldsymbol{\mathcal{A}_\gamma} \odot \boldsymbol{X}, \boldsymbol{\mathcal{A}_\gamma}; \boldsymbol{\theta} \right) - \hat{\ell} \right] \tag{99}$$

$$= \nabla_{\gamma} \mathbb{E} \left[ \log \Phi \left( Y \mid \boldsymbol{\mathcal{A}_\gamma} \odot \boldsymbol{X}, \boldsymbol{\mathcal{A}_\gamma}; \boldsymbol{\theta} \right) \right] - \nabla_{\gamma} \hat{\ell} \tag{100}$$

$$= \nabla_{\gamma} \mathbb{E} \left[ \log \Phi \left( Y \mid \boldsymbol{\mathcal{A}_\gamma} \odot \boldsymbol{X}, \boldsymbol{\mathcal{A}_\gamma}; \boldsymbol{\theta} \right) \right]. \tag{101}$$

Additionally, observe that our gradient estimator requires only forward passes through $\Phi$. Thus, we can use a significantly larger minibatch size when estimating $\nabla_{\gamma} \mathcal{L}$ than when estimating $\nabla_{\boldsymbol{\theta}} \mathcal{L}$ because the intermediate activations do not have to be preserved during forward passes.

**Algorithm 3:** A practical implementation of our adversarial leakage localization algorithm.

**Input:** Training dataset $\mathsf{D} \subset \mathbb{R}^T \times \mathsf{Y}$, initial classifier weights $\boldsymbol{\theta}^{(0)} \in \mathbb{R}^P$, initial unsquashed erasure probabilities $\boldsymbol{\gamma}'^{(0)} \in \mathbb{R}^T$, norm penalty coefficient $\lambda \in \mathbb{R}_+$, log likelihood EMA coefficient $\beta \in [0, 1)$, obfuscator batch size multiplier $M \in \mathbb{Z}_{++}$

**Output:** Trained parameters $\hat{\boldsymbol{\theta}} \in \mathbb{R}^P$, $\hat{\boldsymbol{\gamma}}' \in \mathbb{R}^T$

1 **Function** CalcThetaGrad $(\boldsymbol{x}, y, \boldsymbol{\theta})$
2     $\boldsymbol{\alpha} \sim \mathcal{U}(\{0, 1\}^T)$
3     $l \leftarrow \log \Phi(y \mid \boldsymbol{\alpha} \odot \boldsymbol{x}, \boldsymbol{\alpha}; \boldsymbol{\theta})$
4     $\boldsymbol{g}_c \leftarrow \nabla_{\boldsymbol{\theta}} (-l)$
5     **return** $\boldsymbol{g}_c$

6 **Function** CalcGammaGrad $(\boldsymbol{x}, y, \boldsymbol{\theta}, \boldsymbol{\gamma}, l_{\text{EMA}})$
7     $\boldsymbol{\alpha} \leftarrow (\alpha_t \sim \text{Bernoulli}(1 - \gamma_t) : t = 1, \ldots, T)$
8     $l \leftarrow \log \Phi(y \mid \boldsymbol{\alpha} \odot \boldsymbol{x}, \boldsymbol{\alpha}; \boldsymbol{\theta})$
9     $\nabla_{\boldsymbol{\gamma}'} \left( \frac{1}{2} \|\boldsymbol{\gamma}\|_2^2 \right) \leftarrow \boldsymbol{\gamma} \odot \boldsymbol{\gamma} \odot (\mathbf{1} - \boldsymbol{\gamma})$
10     $\nabla_{\boldsymbol{\gamma}'} l \leftarrow (l - l_{\text{EMA}}) ((\mathbf{1} - \boldsymbol{\alpha}) \odot (\mathbf{1} - \boldsymbol{\gamma}) - \boldsymbol{\alpha} \odot \boldsymbol{\gamma})$
11     $\boldsymbol{g}_o \leftarrow \lambda \nabla_{\boldsymbol{\gamma}'} \left( \frac{1}{2} \|\boldsymbol{\gamma}\|_2^2 \right) + \nabla_{\boldsymbol{\gamma}'} l$
12     $l'_{\text{EMA}} \leftarrow \beta l_{\text{EMA}} + (1 - \beta) l$
13     **return** $\boldsymbol{g}_o, l'_{\text{EMA}}$

14 $t \leftarrow 0$
15 **while** not converged **do**
16     $\mathsf{D}^{(t)} \leftarrow$ SampleMinibatch$(\mathsf{D})$
17     $\mathsf{D}_{\text{aug}}^{(t)} \leftarrow$ AugData(SampleMinibatch$(\mathsf{D})$)
18     $\boldsymbol{\gamma}^{(t)} \leftarrow$ Sigmoid$(\boldsymbol{\gamma}'^{(t)})$
19     $\boldsymbol{g}_c^{(t)} \leftarrow \frac{1}{|\mathsf{D}_{\text{aug}}^{(t)}|} \sum_{(\boldsymbol{x},y) \in \mathsf{D}_{\text{aug}}^{(t)}}$ CalcThetaGrad$(\boldsymbol{x}, y, \boldsymbol{\theta}^{(t)})$
20     $\boldsymbol{\theta}^{(t+1)} \leftarrow$ OptStep$(\boldsymbol{\theta}^{(t)}, \boldsymbol{g}_c^{(t)})$
21     $\boldsymbol{g}_o^{(t)}, l_{\text{EMA}}^{(t+1)} \leftarrow \frac{1}{M|\mathsf{D}^{(t)}|} \sum_{(\boldsymbol{x},y) \in \mathsf{D}^{(t)}} \sum_{m=1}^{M}$ CalcGammaGrad$(\boldsymbol{x}, y, \boldsymbol{\theta}^{(t+1)}, \boldsymbol{\gamma}^{(t)}, l_{\text{EMA}}^{(t)})$
22     $\boldsymbol{\gamma}'^{(t+1)} \leftarrow$ OptStep$(\boldsymbol{\gamma}'^{(t)}, \boldsymbol{g}_o^{(t)})$
23     $t \leftarrow t + 1$
24 **return** $\boldsymbol{\theta}^{(t)}, \boldsymbol{\gamma}'^{(t)}$

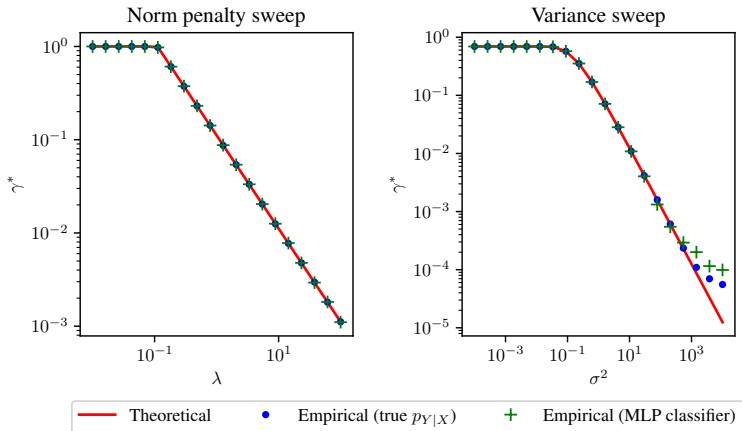

Figure 7: A simple numerical experiment to validate our theoretical results. We run our algorithm on a 2-component Gaussian mixture model where the classifier estimates the conditional distribution of the latent variable based on samples from the mixture. We explicitly compute the optimal erasure probability $\gamma^*$ based on our theoretical analysis, and verify that the results closely match those of our algorithm, both with a multilayer perceptron classifier and with the ground truth conditional latent distribution used in place of the classifier. (left) A plot of $\gamma^*$ while we sweep the $\ell_2$ norm penalty coefficient $\lambda$ with a fixed Gaussian component variance of $\sigma^2 = 1$. For small values of $\lambda$, $\gamma^*$ saturates at 1. (right) A plot of $\gamma^*$ while we sweep $\sigma^2$ and leave $\lambda = 1$. Note that $\gamma^*$ is equal to the mutual information between the latent variable and the samples, which decreases as $\sigma^2$ increases. The mutual information is upper-bounded by the entropy of the latent variable, which is equal to $\log 2 \approx 0.7$ in this case.

## B.6  SIMPLE CORROBORATING NUMERICAL EXPERIMENT

Here we validate our theory with simple numerical experiments where we have explicit expressions for the distributions under consideration and can directly calculate the optimum we expect to converge to.

Consider a simple setting where we have $Y \sim \mathcal{U}\{0,1\}$ and a 1-dimensional power trace $X$ with $p_{X|Y} = \mathcal{N}(Y, \sigma^2)$. In this case, our distributions satisfy

$$p_Y(y) = \tfrac{1}{2}, \tag{102}$$

$$p_{X|Y}(x \mid y) = \tfrac{1}{\sqrt{2\pi\sigma^2}} \exp\left(-\tfrac{1}{2\sigma^2}(x-y)^2\right), \tag{103}$$

$$p_X(x) = \sum_y p_{X|Y}(x \mid y) p_Y(y) = \tfrac{1}{2\sqrt{2\pi\sigma^2}} \left(\exp\left(-\tfrac{1}{2\sigma^2}x^2\right) + \exp\left(-\tfrac{1}{2\sigma^2}(x-1)^2\right)\right), \tag{104}$$

$$p_{Y|X}(y \mid x) = \tfrac{p_{X|Y}(x|y)p_Y(y)}{p_X(x)} = \frac{\exp\left(-\tfrac{1}{2\sigma^2}(x-y)^2\right)}{\exp\left(-\tfrac{1}{2\sigma^2}x^2\right) + \exp\left(\tfrac{1}{2\sigma^2}(x-1)^2\right)}. \tag{105}$$

Based on Proposition 1, we expect the erasure probability to converge to

$$\gamma^* = \min\left\{\frac{\mathbb{I}[Y;X]}{\lambda}, 1\right\}. \tag{106}$$

This is true regardless of our distributions whenever the power trace is 1-dimensional. While in general the mutual information depends on distributions which are not known *a priori* and must be learned from data, in our particular case of a Gaussian mixture model we can directly estimate

$$\mathbb{I}[Y;X] = \sum_{y \in \{0,1\}} \int_{\mathbb{R}} p_Y(y) p_{X|Y}(x \mid y) \left(\log p_{X|Y}(x \mid y) - \log p_X(x)\right) \, dx. \tag{107}$$

We estimate the above quantity using `scipy.integrate.quad` with its default settings as of scipy version 1.14.1, and use it to compute theoretical values for our final obfuscation weight. We train obfuscation weights using the methodology described in previous sections. We run experiments using a multilayer perceptron with 512 hidden units and a ReLU activation as our classifier, as well as using the known $p_{Y|X}$ in place of the classifier. Both our obfuscation weights and multilayer perceptron weights are optimized using `torch.optim.Adam` with `lr=2e-4, betas=(0.9, 0.999), eps=1e-8`. We train for $80\,000$ minibatches with $8192$ examples per minibatch and all examples sampled independently from $p_{X|Y}p_Y$. We exponential moving average our log-likelihood estimate when computing gradients for the obfuscation weights, with the update rule $x_{\text{ema}}^{(t)} = 0.9 \cdot x_{\text{ema}}^{(t-1)} + 0.1 \cdot x^{(t)}$.

## C EXPERIMENTAL DETAILS

### C.1 EXPERIMENTS ON SYNTHETIC DATASETS

Here we present experiments done on synthetic datasets of power traces and associated AES keys. Given that recorded power trace datasets lack 'ground truth' labels about which timesteps are leaking, these experiments are an invaluable validation that the output of our technique is reasonable. Additionally, synthetic datasets allow us to test the limits of our algorithm by varying data generation parameters to extreme values. Refer to our code for further details: in the supplementary materials.

#### C.1.1 DATA GENERATION PROCEDURE

We base our synthetic power trace datasets on the Hamming weight leakage model of Mangard et al. (2007, ch. 4). This model[3] assumes that we have a device which executes a cryptographic algorithm as a sequence of operations on data. As above, let $\boldsymbol{X} := (X_t : t = 1, \ldots, T)$ be a random vector with range $\mathbb{R}^T$ which encodes power consumption. Let $\boldsymbol{D} := (D_t : t = 1, \ldots, T)$ and $\boldsymbol{O} := (O_t : t = 1, \ldots, T)$ be random vectors denoting the data and operations, respectively, where each $D_t$ has range $\{0,1\}^{n_{\text{bits}}}$ (i.e. a sequence of $n_{\text{bits}}$ bits) and each $O_t$ has range $[1 .. n_{\text{ops}}]$ for some $n_{\text{bits}}, n_{\text{ops}} \in \mathbb{Z}_{++}$. For each $t \in [1 .. T]$, we can decompose

$$X_t = X_{\text{data},t} + X_{\text{op},t} + X_{\text{resid},t} \tag{108}$$

with dependency structure illustrated in the causal diagram of figure 8. Note that $X_{\text{data},t}$ is directly associated with the data $D_t$, $X_{\text{op},t}$ is directly associated with the operation $O_t$, and $X_{\text{resid},t}$ captures the randomness in power consumption we would see if we were to repeatedly measure power consumption with a fixed operation and data (e.g. due to other processes on the device independently of the encryption process, or noise due to the thermal motion of electrons in wires).

The authors of Mangard et al. (2007) experimentally characterize the power consumption of a cryptographic device and find that it is reasonable to approximate $X_{\text{data},t}$ as Gaussian noise with $D_t$-dependent mean, $X_{\text{op},t}$ as Gaussian noise with $O_t$-dependent mean, and $X_{\text{resid},t}$ as Gaussian noise with a constant mean (which we will assume to be $0$). For their device, the mean of $X_{\text{data},t}$ is proportional to

$$n_{\text{bits}} - \text{HammingWeight}(D_t) := \sum_{k=1}^{n_{\text{bits}}} 1 - D_{t,k}, \tag{109}$$

i.e. the number of bits of $D_t$ which are equal to $0$. Additionally, the per-$O_t$ means of $X_{\text{op},t}$ are approximately Gaussian-distributed.

We adopt these approximations for our experiments, though we emphasize that they are not universally-applicable to cryptographic devices. For example, the Hamming weight dependence of the mean of $X_{\text{data},t}$ on $D_t$ is due to the fact that their device 'pre-charges' all of its data bus lines to $1$, then drains the charge from the lines which should represent $0$, thereby consuming power proportional to the number of lines which represent $0$. Many devices operate differently. Additionally, cryptographic hardware is often explicitly designed to obfuscate the association between power consumption and data/operations as a defense mechanism against side-channel attacks.

---

[3]Mangard et al. (2007) uses the notation $P_{\text{total}} = P_{\text{op}} + P_{\text{data}} + P_{\text{el. noise}} + P_{\text{const}}$. For clarity and consistency, we alter the notation, consolidate $P_{\text{el. noise}}$ and $P_{\text{data}}$ into a single variable, and more-explicitly define the probabilistic nature of the variables and the associations between them.

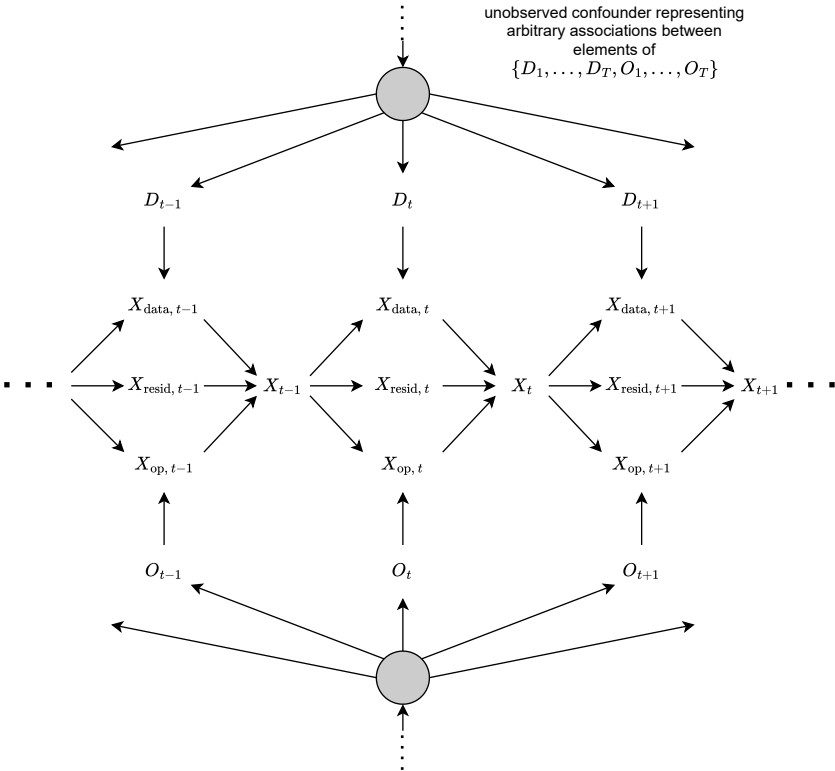

Figure 8: Causal diagram which shows the independence conditions between different components of power consumption which we have assumed in our synthetic power trace dataset. Power consumption at a time $t$ is decomposed as $X_t = X_{\text{data},t} + X_{\text{op},t} + X_{\text{resid},t}$ where $X_{\text{data},t}$ is directly associated with the present data $D_t$, $X_{\text{op},t}$ is directly associated with the present operation $O_t$, and $X_{\text{resid},t}$ is directly associated with neither, and accounts for all sources of randomness in power consumption not directly associated with the data or operation. We assume that arbitrary associations may exist between the data and operations at different points in time. We assume that power consumption at time $t + 1$ is associated with that at time $t$ due to the 'inertia' of power consumption.

**Algorithm 4:** Simplified procedure for generating synthetic power trace datasets based on the Hamming weight leakage model of Mangard et al. (2007).

**Input:**    Dataset size $N \in \mathbb{Z}_{++}$,
              Timesteps per power trace $T \in \mathbb{Z}_{++}$,
              Bit count $n_{\text{bits}} \in \mathbb{Z}_{++}$,
              Operation count $n_{\text{ops}} \in \mathbb{Z}_{++}$,
              1st-order leaking timestep count $n_{\text{lkg}} \in \mathbb{Z}_{+}$,
              Data-dependent noise variance $\sigma_{\text{data}}^2 \in \mathbb{R}_{+}$,
              Operation-dependent noise variance $\sigma_{\text{op}}^2 \in \mathbb{R}_{+}$,
              Residual noise variance $\sigma_{\text{resid}}^2 \in \mathbb{R}_{+}$
**Output:** Synthetic dataset $\mathsf{D} \subset \mathbb{R}^T \times [1 \mathinner{\ldotp\ldotp} 2^{n_{\text{bits}}}]$

1   $\{k^{(n)} : n \in [1 \mathinner{\ldotp\ldotp} N]\} \overset{\text{i.i.d.}}{\sim} \mathcal{U}(\{0,1\}^{n_{\text{bits}}})$                           `// cryptographic keys`

2   $\{w^{(n)} : n \in [1 \mathinner{\ldotp\ldotp} N]\} \overset{\text{i.i.d.}}{\sim} \mathcal{U}(\{0,1\}^{n_{\text{bits}}})$                               `// plaintexts`

3   $\{o_t : t \in [1 \mathinner{\ldotp\ldotp} T]\} \overset{\text{i.i.d.}}{\sim} \mathcal{U}([1 \mathinner{\ldotp\ldotp} n_{\text{ops}}])$                               `// operations`

4   $\{\tilde{x}_{\text{op},o} : o \in [1 \mathinner{\ldotp\ldotp} n_{\text{ops}}]\} \overset{\text{i.i.d.}}{\sim} \mathcal{N}(0, \sigma_{\text{op}}^2)$          `// per-operation power consumption`

5   $\boldsymbol{x}_{\text{op}} \leftarrow (\tilde{x}_{\text{op},o_t} : t = 1, \ldots, T)$         `// operation-dependent power consumption`

6   $\mathsf{T}_{\text{lkg}} \sim \mathcal{U}\left(\dbinom{[1 \mathinner{\ldotp\ldotp} T]}{n_{\text{lkg}}}\right)$         `// leaking timesteps, sampled w/o replacement`

7 **for** $n \in [1 \mathinner{\ldotp\ldotp} N]$ **do**

8     $y^{(n)} \leftarrow \text{AES-SBOX}(k^{(n)} \oplus w^{(n)})$                 `// sensitive variable`

9     $\boldsymbol{x}_{\text{resid}}^{(n)} \sim \mathcal{N}_T(\boldsymbol{0}, \sigma_{\text{resid}}^2 \boldsymbol{I})$              `// residual power consumption`

10    **for** $t \in \mathsf{T}_{lkg}$ **do**

11      $d_t^{(n)} \leftarrow y^{(n)}$        `// timesteps at which the sensitive variable leaks`

12    **for** $t \in [1 \mathinner{\ldotp\ldotp} T] \setminus \mathsf{T}_{\text{lkg}}$ **do**

13      $d_t^{(n)} \sim \mathcal{U}(\{0,1\}^{n_{\text{bits}}})$         `// other data which we treat as random`

14    **for** $t \in [1 \mathinner{\ldotp\ldotp} T]$ **do**

15      $x_{\text{data},t}^{(n)} \leftarrow \sigma_{\text{data}}(4 - \text{HammingWeight}(d_t^{(n)}))/\sqrt{2}$    `// data-dependent power`
               `consumption`

16    $\boldsymbol{x}^{(n)} \leftarrow \boldsymbol{x}_{\text{data}}^{(n)} + \boldsymbol{x}_{\text{op}} + \boldsymbol{x}_{\text{resid}}^{(n)}$            `// total power consumption`

17 **return** $\left\{(\boldsymbol{x}^{(n)}, y^{(n)}) : n \in [1 \mathinner{\ldotp\ldotp} N]\right\}$

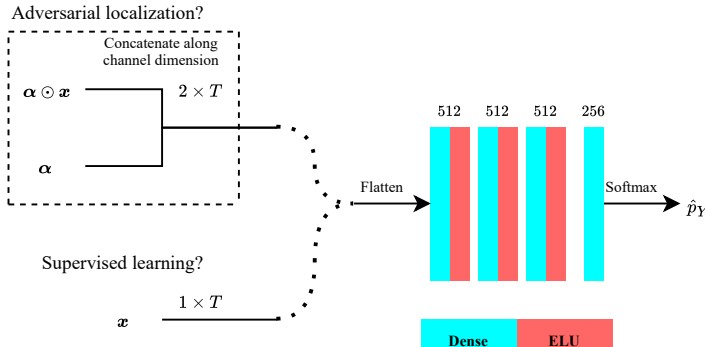

Figure 9: Multilayer perceptron architecture used for our experiments on synthetic datasets. We use nearly the same architecture for adversarial and supervised experiments. For adversarial experiments we feed the MLP both a noisy power trace and the noise vector, concatenated along the channel dimension. For supervised experiments (e.g. for computing neural network 'interpretability' baselines), we simply feed the power trace to the MLP.

| Countermeasure | Training steps | $\boldsymbol{\theta}$ LR | $\boldsymbol{\gamma}'$ LR |
|---|---|---|---|
| Unprotected | $5 \times 10^4$ | $10^{-3}$ | $10^{-3}$ |
| Random delays | $5 \times 10^4$ | $10^{-3}$ | $10^{-3}$ |
| Random shuffling | $5 \times 10^4$ | $10^{-4}$ | $10^{-3}$ |
| Boolean masking | $2 \times 10^5$ | $2 \times 10^{-4}$ | $10^{-3}$ |

Table 2: Hyperparameters used in our synthetic data experiments.

A simplified version of our data generation procedure is shown in algorithm 4. In our experiments we have also simulated desynchronization, Boolean masking and shuffling countermeasures. We have also explored low-pass filtering (exponentially-weighted moving averaging) the traces as a rough approximation to the 'inertia' of power consumption in real devices. We omit these details for brevity in the above algorithm, but they can be found in our code in the supplementary materials.

### C.1.2 NEURAL NET ARCHITECTURE AND HYPERPARAMETERS

For all synthetic data experiments we use the simple multilayer perceptron architecture shown in figure 9. To avoid potential overfitting effects we generate data online for these experiments. Note that overfitting is a factor in our experiments on the finite-size real datasets. We use the dataset settings $N = \infty$, $T = 500$, $n_{\text{bits}} = 8$, $n_{\text{ops}} = 32$, and $\sigma_{\text{data}}^2 = \sigma_{\text{resid}} = \sigma_{\text{op}} = 1.0$. To simulate the inertia of power rails of real hardware, we apply an exponentially-weighted moving average $x_{\text{EMA}}^{(t)} \leftarrow 0.9 \cdot x_{\text{EMA}}^{(t-1)} + x^{(t)}$.

For all experiments, both $\boldsymbol{\gamma}'$ and $\boldsymbol{\theta}$ are trained with the AdamW optimizer with $\beta_1 = 0.9$, $\beta_2 = 0.999$, and weight decay disabled. We use a minibatch size of 1024 for $\boldsymbol{\theta}$ and $8 \times 1024$ for $\boldsymbol{\gamma}'$ (i.e. $M = 8$ in algorithm 3). Our log-likelihood EMA coefficient is set to $\beta = 0.9$. The experiment-dependent hyperparameters are shown in table 2.

### C.2 EXPERIMENTS ON RECORDED POWER TRACE DATASETS

Real-world associations between AES keys and power traces are complicated and are not completely characterized by our simple synthetic data generation process. Thus, evaluation on datasets recorded from real cryptographic hardware is critical. Here we present experiments done on a variety of publicly-available power trace datasets which are commonly used in the literature of deep learning-based side channel attacks.

### C.2.1 EVALUATING PERFORMANCE WITHOUT GROUND TRUTH KNOWLEDGE ABOUT LEAKAGE

For synthetic datasets it is easy to tell whether the output of our algorithm is reasonable because we know the points in time at which power consumption is directly affected by sensitive variables.

However, we lack this knowledge for real datasets and in our motivating setting, which is hardware designers seeking to understand why their cryptographic implementation leaks. Intuitively, a power measurement is 'more-leaky' if it is more-useful for performing a side-channel attack. Based on this intuition, we propose a novel performance metric based on the extent to which the leakage value assigned to a power measurement positively correlates with the performance gain of a profiled side-channel attack on partial power traces which includes this measurement. This metric is similar to the evaluation methods of Masure et al. (2019) and Hettwer et al. (2020), while overcoming limitations of both.

In Masure et al. (2019), the authors propose the Gradient Visualization techinque and evaluate its efficacy by performing a Gaussian template attack on the 'most-leaky' points it has identified, with better template attack performance indicating that the identified points were 'leakier'. The limitation of this approach is that while it is sensitive to true-positive leakage detection, it is not sensitive to false-negatives.

In Hettwer et al. (2020), the authors evaluate several existing neural network attribution techniques as ways to localize leakage. They propose several performance metrics which are all conceptually-similar to the following: 1) Train a deep neural network to perform a side-channel attack. 2) Successively replace individual timesteps with random noise in order of their estimated 'leakiness'. 3) Visually assess how the performance of the neural net changes as points are replaced. It is expected that performance will decrease monotonically as points are replaced, and a steeper negative slope early on indicates that 'leakier' points have been identified. The limitations of this approach are that it requires visual inspection of the curves, and that because the same style of neural net is used for both leakage localization and evaluation of the leakage assessment, there is a risk that evaluations are biased towards types of leakage which happen to be useful to that style of neural net.

To evaluate the performance of a leakage assessment, we sort the timesteps based on their estimated leakage and partition the sorted points into sets of 10. We fit a Gaussian mixture model (similar to a Gaussian template attack) to each of these sets of 10 measurements and fit a Gaussian distribution to the rank of the correct key for each power trace in the conditional distribution returned by the model. We then compute a 'softened' version of the Kendall $\tau$ rank correlation coefficient between the amount of leakage estimated for a set of points and the mean rank assigned to the correct key. It is 'softened' in the sense that we compute the expected value of the difference between concordant and discordant pair counts. See algorithm 5 for details. We use the Kendall $\tau$ correlation coefficient because it checks only the monotonicity of a curve, rather than its shape. We model the performance metrics as random variables because there are generally many performance evaluations which are statistically-indistinguishable but may differ due to random change, and expectation is a natural way to 'downweight' the pairs of measurements for which this the case.

### C.2.2 NEURAL NET ARCHITECTURE AND HYPERPARAMETERS

For all 'real dataset' experiments we use a simple VGG-like CNN (Simonyan & Zisserman, 2015), similarly to Benadjila et al. (2020); Zaid et al. (2020) (see figure 10). As done in the publicly-available implementation of Wouters et al. (2020), we initialize the dense layers with `torch.nn.init.xavier_uniform_` and the convolutional layers with `torch.nn.init.kaiming_uniform_`. We find that this detail is critical, as networks completely fail to generalize on some datasets when we use the default PyTorch initializations.

For all experiments, both $\gamma'$ and $\theta$ are trained with the AdamW optimizer with $\beta_1 = 0.9$ and $\beta_2 = 0.999$. We set the algorithm 3 hyperparameters to $M = 8$ and $\beta = 0.9$. To tune hyperparameters, we first tried learning rates in $\{1 \times 10^{-6}, 2 \times 10^{-6}, \ldots, 7 \times 10^{-6}\}$ for DPAv4 and $\{1 \times 10^{-6}, \ldots, 9 \times 10^{-6}\} \cup \{1 \times 10^{-5}, \ldots, 9 \times 10^{-5}\} \cup \{10^{-4}\}$ for ASCADv1. For each of these models we tested weight decay values of 0 and $10^{-2}$, and data augmentation via additive Gaussian input noise with standard deviation values of 0 and 0.25. We then selected the hyperparameters which minimized the mean correct-key rank on our validation dataset after training for $10^4$ steps. This sweep is displayed in figure 11 for DPAv4 and in figure 12 for ASCADv1.

For the neural net interpretation baselines, we trained a neural network with these optimal hyperparameters for $10^4$ steps and early-stopped based on validation rank. For the adversarial leakage localization experiments, we used these settings for our classifier and tuned $\lambda$ and the learning rate of $\gamma'$ *ad hoc*.

**Algorithm 5:** Our metric for evaluating the fidelity of a leakage localization attempt. We call this metric the Gaussian Mixture Model Performance Rank Correlation (GMM-PRC).

**Input:** Profiling dataset $\mathsf{D} := \{(\boldsymbol{x}^{(n)}, y^{(n)}) : n \in [1 \,..\, N]\} \subset \mathbb{R}^T \times \mathsf{Y}$, attack dataset
$\quad\quad \mathsf{D}_\mathrm{a} := \{(\boldsymbol{x}_\mathrm{a}^{(n)}, y_\mathrm{a}^{(n)}) : n \in [1 \,..\, N_\mathrm{a}]\} \subset \mathbb{R}^T \times \mathsf{Y}$, leakage assessment $\boldsymbol{\gamma} \in \mathbb{R}^T$, order of
$\quad\quad$ GMM attack $m \in \mathbb{Z}_{++}$.
**Output:** Gaussian mixture model performance rank correlation (GMM-PRC) $\tau \in [-1, 1]$.

**1 Function** PerformGMMAttack $(\boldsymbol{t} \in [1 \,..\, T]^m)$
$\quad$ // Fit a Gaussian mixture model to the profiling dataset.
**2** $\quad$ **for** $y \in \mathsf{Y}$ **do**
**3** $\quad\quad$ $\mathsf{D}_y \leftarrow \{(\boldsymbol{x}^{(n)}, y^{(n)}) : n \in [1 \,..\, N] : y^{(n)} = y\}$
**4** $\quad\quad$ $N_y \leftarrow |\mathsf{D}_y|$
**5** $\quad\quad$ $\boldsymbol{\mu}_y \leftarrow \frac{1}{N_y} \sum_{(\boldsymbol{x}, y) \in \mathsf{D}_y} \boldsymbol{x_t}$
**6** $\quad\quad$ $\boldsymbol{\Sigma}_y \leftarrow \frac{1}{N_y - 1} \sum_{(\boldsymbol{x}, y) \in \mathsf{D}_y} (\boldsymbol{x_t} - \boldsymbol{\mu}_y)(\boldsymbol{x_t} - \boldsymbol{\mu}_y)^\top$

$\quad$ // Compute the rank of the correct intermediate variable for each trace in
$\quad\quad$ the attack dataset
**7** $\quad$ **for** $n = 1, \ldots, N_a$ **do**
**8** $\quad\quad$ **for** $y \in \mathsf{Y}$ **do**
**9** $\quad\quad\quad$ $u_y^{(n)} \leftarrow \log \mathcal{N}(\boldsymbol{x}_{\mathrm{a}, \boldsymbol{t}}^{(n)}; \boldsymbol{\mu}_y, \boldsymbol{\Sigma}_y) + \log N_y$
**10** $\quad\quad$ $r^{(n)} \leftarrow |\{y \in \mathsf{Y} : u_y^{(n)} \geq u_{y^{(n)}}^{(n)}\}|$

$\quad$ // Compute the mean and standard deviation of the rank
**11** $\quad$ $\mu \leftarrow \frac{1}{N_\mathrm{a}} \sum_{n=1}^{N_\mathrm{a}} r^{(n)}$
**12** $\quad$ $\sigma^2 \leftarrow \frac{1}{N_{\mathrm{a}-1}} \sum_{n=1}^{N_\mathrm{a}} (r^{(n)} - \mu)^2$
**13** $\quad$ **return** $\mu, \sigma^2$

// Compute the statistics of the rank of the correct intermediate variable
$\quad$ for groups of timesteps with similar relative amount of leakage, as
$\quad$ estimated by the leakage localization attempt we are evaluating.
**14** $\boldsymbol{\kappa} \leftarrow$ ArgSort$(\boldsymbol{\gamma})$
**15 for** $l = 0, \ldots, \lfloor \frac{T}{m} \rfloor - 1$ **do**
**16** $\quad$ $\boldsymbol{t}_l \leftarrow \boldsymbol{\kappa}_{(m \cdot l, \ldots, m \cdot (l+1) - 1)}$
**17** $\quad$ $\mu_l, \sigma_l^2 \leftarrow$ PerformGMMAttack$(\boldsymbol{t}_l)$

// Compute the expected value of the Kendall $\tau$ rank correlation coefficient
$\quad$ of these GMM performance statistics.
**18** $\tau \leftarrow 0$
**19 for** $i = 1, \ldots, \lfloor \frac{T}{m} \rfloor$ **do**
**20** $\quad$ **for** $j = i + 1, \ldots, \lfloor \frac{T}{m} \rfloor$ **do**
**21** $\quad\quad$ $\tau \leftarrow \tau + \int_0^\infty \mathcal{N}(x; \mu_i - \mu_j, \sigma_i^2 + \sigma_j^2)\, dx$
**22** $\tau \leftarrow \frac{2\tau}{\lfloor \frac{T}{m} \rfloor^2 + \lfloor \frac{T}{m} \rfloor}$
**23 return** $\tau$

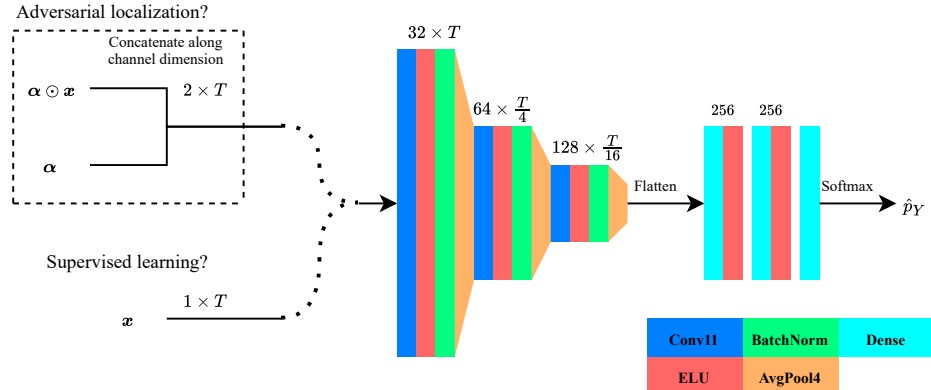

Figure 10: Simple VGG-style CNN architecture used for our experiments on recorded power trace datasets. We use nearly the same architecture for adversarial and supervised experiments. For adversarial experiments we feed the CNN both a noisy power trace and the noise vector, concatenated along the channel dimension. For supervised experiments (e.g. for computing neural network 'interpretability' baselines), we simply feed the power trace to the CNN.

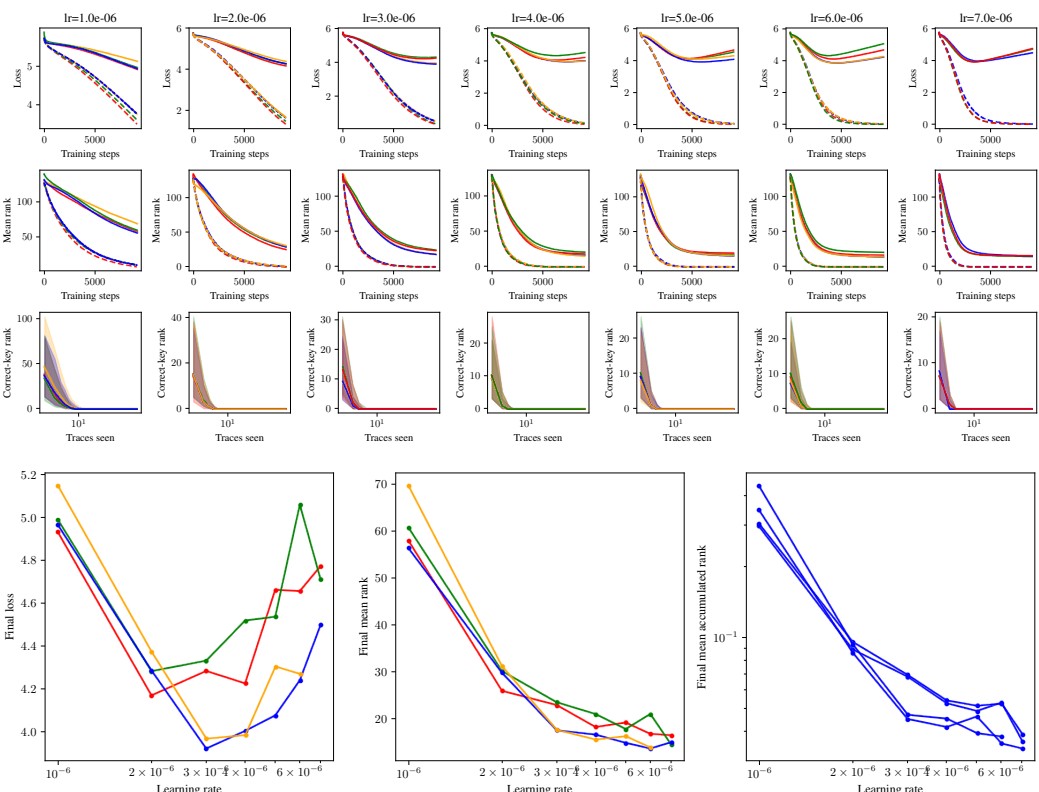

Figure 11: Performance metrics on the DPAv4 dataset for various learning rates.

Figures 13 and 14 provide detailed comparisons of the leakage assessments done by our compared baselines. The bottom rows of these figures show the mean $\pm$ standard deviation of the rank of the correct intermediate variable that is used to compute our GMM-PRC metric. Observe that high GMM-PRC values in table 1 correspond to monotonically-decreasing rank in these plots.

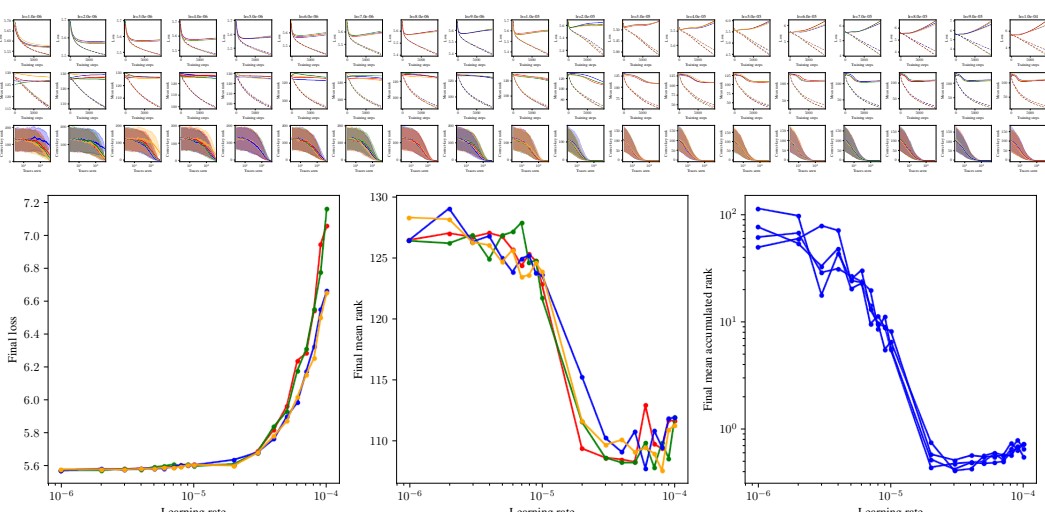

Figure 12: Performance metrics on the ASCADv1-fixed dataset while sweeping the learning rate.

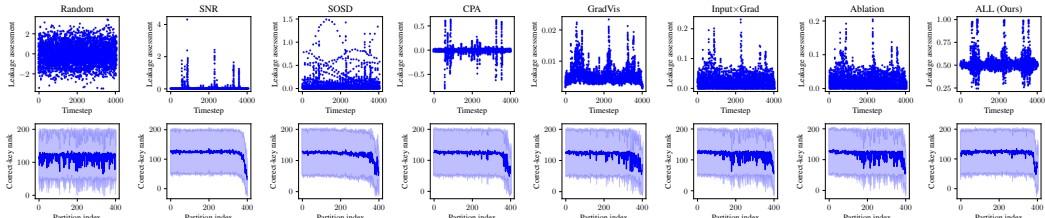

Figure 13: Qualitative comparison of considered methods on the DPAv4 dataset (right column: ours). The top row shows plots of the estimated leakage of $X_t$ vs $t$ for each method. The bottom row shows plots of the performance of a Gaussian mixture model-based side channel attack (lower is better), vs the relative estimated leakage of the partition of timesteps it is trained on (solid: mean, shaded: std. dev.). We expect the correlation to be more negative for more accurate estimated leakage values.

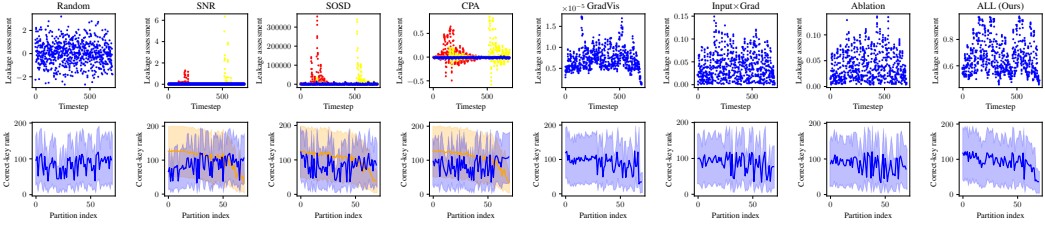

Figure 14: Qualitative comparison of considered methods on the ASCADv1 (fixed key) dataset (right column: ours). The top row shows plots of the estimated leakage of $X_t$ vs $t$ for each method. The bottom row shows plots of the performance of a GMM-based side channel attack (lower is better), vs the relative estimated leakage of the partition of timesteps it is trained on (solid: mean, shaded: std. dev.). SNR, SOSD, and CPA are unable to detect leakage due to the Boolean masking of this dataset, but because this dataset provides unrealistic knowledge of the random mask values, as an idealistic baseline we show the results from targeting the mask (red) and masked sensitive variable (yellow) separately. The orange traces in the bottom row correspond to summing these leakage assessments.