# OpenReview forum: "Learning to localize leakage of cryptographic keys through power consumption"
_ICLR.cc/2025/Conference — ICLR 2025 Conference Withdrawn Submission_

### Official Review · Reviewer_2VYq · 2024-10-25

**Soundness:** 2
**Presentation:** 3
**Contribution:** 2
**Rating:** 3
**Confidence:** 4

**Summary:**

This work involves another occlusion method that utilizes an information-theoretical approach to identify occlusion points.
This method is called Adversarial Leakage Localization (ALL). This approach offers an alternative explainability technique for deep neural networks within the context of profiling side channel attack.

**Strengths:**

The work consists of an automated method to find multiple masks of occlusion. Its strength lies in utilizing information theory to model the problem as an optimization problem, which introduces erasure probability to identify not just one mask, but multiple masks.

**Weaknesses:**

There are multiple weakness in this work.
1. Firstly, the technique requires tuning the hyperparameter \lambda which can be challenging and may not always yield optimal results without proper calibration.
2. Furthermore, the Points of Interest (PoIs) selected by the mask may not directly contribute to key recovery, requiring additional investigation. Specifically, it remains to be verified whether applying the mask effectively reduces the guessing entropy to zero when used in conjunction with a deep neural network. If not, the mask may provide misleading insights.
3. Furthermore, it is not the first time occlusion has been used in profiling side-channel analysis. Two very important works are not cited: [1] and [2]. Notably, the current work bears significant similarities to [1], making a comparative analysis essential to contextualize the contributions.
4. Additionally, the limited dataset selection raises concerns. Specifically, DPAv4 has known flaws in its masking schemes [3], rendering it a suboptimal choice. To enhance the robustness of the results, testing on ASCADv1 (random key setting) and other datasets like AES_HD or DPAv4.2 is recommended.

5. The paper would benefit from prominently featuring the C2.1 metric in the main text. Additionally, a clear definition of the C2.1 metric is necessary to facilitate understanding.
6. The abstract inaccurately suggests a defense methodology is proposed. Instead, the paper presents a model-agnostic explainability technique for deep neural networks. To avoid confusion, rephrase the abstract to reflect the correct focus on explainability.

**Questions:**

It would be intriguing to investigate whether applying various masks for feature selection, as in [1], affects the effectiveness of the proposed method's mask. Specifically, examining if the Template Attack (i.e Gaussian template) remains successful with features selected by the proposed mask would provide valuable insights.

---

> ### Author Response · Authors · 2024-11-12
>
> Thank you for your useful comments. It appears that you did not state which papers [1, 2, 3] refer to -- could you please do so in order for us to better address your concerns?

---

### Official Review · Reviewer_ReRp · 2024-10-28

**Soundness:** 1
**Presentation:** 3
**Contribution:** 1
**Rating:** 3
**Confidence:** 5

**Summary:**

This paper presents an intriguing approach leveraging a machine learning-based classifier to identify portions of software code that leak more sensitive information than others, using the Advanced Encryption Standard (AES) as a case study.  Specifically, the method ranks code segments based on their relative likelihood of leaking a predefined secret—the AES key. The approach is validated on synthetic AES power trace data generated according to the Hamming weight leakage model, an open-source dataset.

**Strengths:**

This method is evaluated using first-order statistics and neural network interpretability techniques, marking a novel application of machine learning in side-channel analysis.

The reviewer found this paper notable for its innovative use of advanced machine learning techniques in the hardware security domain. However, the following key concerns warrant further attention:

**Weaknesses:**

Threat Model: What is the threat model assumed in this work? In hardware security research, particularly in side-channel attacks targeting AES key extraction, attackers are typically assumed to have strong capabilities, often operating in a white-box setup. This implies they can partition the victim’s software program and establish checkpoints for periodic power trace collection. Does this paper assume a similar setup?

Applicability Across Devices: Assuming a white-box setup, a natural follow-up question is: how transferable are the collected traces to other devices? In power side-channel attacks, traces collected from one specific device often cannot be directly applied to another due to inter-device variations, as discussed in prior work [1]. If an attacker has access to only one device, they must generally recollect traces for each new target device, unless they have complete device-specific knowledge (see Point 1 above). This raises an important question for the proposed defensive solution: is it universally effective across different device setups, or does it require extensive adaptation?

Hardware Platform Variability: AES is widely deployed across a range of hardware platforms, including MCU, FPGA, GPU, and CPU, each with unique programming and operational characteristics. Given the objective of securing the most vulnerable or leaky portions of code, it would be helpful to discuss whether the proposed solution can generalize across these varied hardware platforms. How feasible is it for this approach to offer a uniform level of security given these differences?

Comparison with Existing Baselines: This paper would benefit from a comparative analysis with existing baseline methods. Numerous techniques exist from hardware, algorithmic, and cryptographic perspectives to mitigate AES key leakage in side-channel attacks. A comparison would help establish the proposed method’s effectiveness and distinguish it from alternative solutions.

Reference:
[1] Omar Choudary and Markus G. Kuhn. 2014. Template Attacks on Different Devices. Springer International Publishing, Cham, 179–198.

**Questions:**

see above weaknesses.

---

### Official Review · Reviewer_2XwM · 2024-11-03

**Soundness:** 4
**Presentation:** 4
**Contribution:** 3
**Rating:** 8
**Confidence:** 3

**Summary:**

This paper focusses on the physical implementations of cryptographic algorithms and the property that they leak sensitive information such as cryptographic keys.  The form of leakage central to the subject matter of this paper is power consumption over time (temporal) and cryptographic algorithm studied is AES.

The paper proposes a novel deep learning algorithm by formulating an adversarial game between played between a classifier trained to the conditional distribution a key given power measurements and an obfuscator which probabilistically erases individual power measurements and is trained to minimize the classifier estimated log-likelihood of the correct key.  Both a theoretical characterization and an empirical demonstration of the efficacy of the proposed algorithm on real and synthetic datasets of power measurements from implementations of the AES cryptographic standard.

**Strengths:**

A novel deep learning-based power side-channel leakage localization algorithms is thoroughly set within the context of previous literature and techniques.  The theoretical characterization is comprehensively presented in Appendix B.  A performance comparison of the proposed algorithm is presented in Table 1 with a necessary elaboration presented in Appendix C Experimental Details.
Of special interest is the comparison of the author's method to the known Gradient Visualization technique.

The work is original and of high quality.  It draws at times for inspiration of known techniques such as GANs [Goodfellow 2014] and ENCO algorithm [Lippe 2022] but the author's algorithm is distinct.

**Weaknesses:**

Acknowledging that this problem space is complex a few items would benefit with further explaination.

The proposed algorithm is stated as being highly sensitive to the choice of "lambda".  How would "lambda" be chosen in practise with a given recorded AES power trace - grid search?.   Does the power trace have to be normalized in some way as a precursor to establishing a value for "lambda".

In the first paragraph of section A.3, I would replace "imperfect manufacturing processes" with "PVT variations"  a more standard term, familiar to digital designers (PVT==(Process, Voltage, Temperature).

The assumption that pre-charged busses lead to the claimed results is valid.  However, pre-charged buses in the digital realization may apply to only a limited number of AES realizations and thus by implication a reduced scope of impact.

**Questions:**

How are the relative and absolute values of the noise variances chosen?  (Algorithm 4)

A more comprehensive discussion on True-Positive Leakage Detection vs False-Negative Leakage Detection for the proposed algorithm.
Additional comments on this topic would be valuable.

---

### Official Review · Reviewer_uset · 2024-11-03

**Soundness:** 2
**Presentation:** 3
**Contribution:** 2
**Rating:** 5
**Confidence:** 4

**Summary:**

The authors propose a reinforcement method to enhance the security of AES encryption against side-channel attacks by identifying key leakage points. This approach uses a classifier-obfuscator framework: the classifier attempts to predict sensitive intermediate variables from power traces, while the obfuscator selectively erases sections of the trace to hinder the classifier’s accuracy. By theoretically and experimentally pinpointing the time points that leak the most information, this method enables designers to focus security improvements on these critical areas.

**Strengths:**

Clear and thorough discussion of previous work and related research.

**Weaknesses:**

The experiments focus solely on defending against AES power leakage. However, it would be valuable to explore broader applications of this defense method, as AES already benefits from various existing techniques, such as MPC-based methods and S-box replacement. Demonstrating the potential of this defense for other encryption schemes or algorithmic leakages could further enhance the paper’s contribution.
One AES demonstration cannot prove the authors' claim that “Our algorithm is generic enough to be applicable to other cryptographic standards as well.”

Power traces differ across hardware platforms, raising questions about the model’s transferability. A model trained on one device may not perform effectively on another due to variations in leakage characteristics.
Does this imply that for each device, we would need to rebuild the model and collect a large number of traces to create a new dataset?

Additionally, the authors suggest that this method can help designers identify leakage points; however, what is more crucial is how to eliminate the leakage. On this point, the paper does not provide an effective solution.

**Questions:**

Could the authors please provide an analysis and demonstration of the model’s transferability across different devices and various AES chips or implementations?

---

### Author Response · Authors · 2024-11-28

We appreciate the useful feedback we have received. Reviewers requested extensive additional experiments and evaluation strategies, which we believe will significantly strengthen our paper. However, there was insufficient time during the rebuttal period to complete these in a careful and rigorous manner. Thus, we have chosen to withdraw our paper and resubmit to a future conference so that we have adequate time to address this feedback.

Reviewer 2VQy: We would appreciate it if you could tell us which papers [1, 2, 3] refer to in your review.

---

### Note · Authors · 2024-11-28

I have read and agree with the venue's withdrawal policy on behalf of myself and my co-authors.